# Structure of photosystem I-LHCI-LHCII from the green alga *Chlamydomonas reinhardtii* in State 2

Zihui Huang[1,7], Liangliang Shen [2,3,7], Wenda Wang [2], Zhiyuan Mao[2,3], Xiaohan Yi[1], Tingyun Kuang [2], Jian-Ren Shen [2,4✉], Xing Zhang [1,5,6✉] & Guangye Han [2✉]

Photosystem I (PSI) and II (PSII) balance their light energy distribution absorbed by their light-harvesting complexes (LHCs) through state transition to maintain the maximum photosynthetic performance and to avoid photodamage. In state 2, a part of LHCII moves to PSI, forming a PSI-LHCI-LHCII supercomplex. The green alga *Chlamydomonas reinhardtii* exhibits state transition to a far larger extent than higher plants. Here we report the cryo-electron microscopy structure of a PSI-LHCI-LHCII supercomplex in state 2 from *C. reinhardtii* at 3.42 Å resolution. The result reveals that the PSI-LHCI-LHCII of *C. reinhardtii* binds two LHCII trimers in addition to ten LHCI subunits. The PSI core subunits PsaO and PsaH, which were missed or not well-resolved in previous Cr-PSI-LHCI structures, are observed. The present results reveal the organization and assembly of PSI core subunits, LHCI and LHCII, pigment arrangement, and possible pathways of energy transfer from peripheral antennae to the PSI core.

[1] Department of Biophysics, and Department of Pathology of Sir Run Run Shaw Hospital, Zhejiang University School of Medicine, Hangzhou, Zhejiang, China. [2] Photosynthesis Research Center, Key Laboratory of Photobiology, Institute of Botany, Chinese Academy of Sciences, Beijing, China. [3] University of Chinese Academy of Science, Beijing, China. [4] Institute for Interdisciplinary Science, and Graduate School of Natural Science and Technology, Okayama University, Okayama, Japan. [5] Center of Cryo-Electron Microscopy, Zhejiang University School of Medicine, Hangzhou, Zhejiang, China. [6] Zhejiang Laboratory for System and Precision Medicine, Zhejiang University Medical Center, Hangzhou, Zhejiang, China. [7] These authors contributed equally: Zihui Huang, Liangliang Shen. ✉email: shen@cc.okayama-u.ac.jp; xzhang1999@zju.edu.cn; hanguangye@ibcas.ac.cn

Photosystem I and II (PSI and PSII) are two multisubunit pigment-protein complexes embedded in the thylakoid membranes of higher plants, algae, and cyanobacteria[1], and carry out light capture and subsequent photochemical reactions in oxygenic photosynthesis. PSI and PSII each consist of core complexes and its peripheral antenna systems[1,2]. The core complexes of PSI and PSII are highly conserved, whereas the outer antenna systems varies remarkably in different organisms as a result of adaptation to different light conditions[1–4]. In higher plants and green algae, the peripheral antenna system is composed of membrane-spanning proteins encoded by the light-harvesting complex (Lhc) multigenic family[5]. In higher plants, the PSI peripheral antenna system is LHCI containing four Lhca proteins Lhca1-4[6,7], whereas the PSII antenna system consists of mainly six Lhcb proteins (Lhcb1-6) encoded by *Lhcb1-6* genes. Lhcb1-3 organize as homo- or hetero-trimers to form the major LHCII[8,9]. The other chlorophyll (Chl) *a/b*-binding proteins, Lhcb4, Lhcb5, and Lhcb6, also called CP29, CP26, and CP24, exist as monomeric antennae[10,11]. In the green alga *Chlamydomonas reinhardtii* (*C. reinhardtii*), a unicellular green alga used as a model organism for photosynthesis studies, PSI antenna size is far larger than that of higher plants and ten LHCI subunits encoded by *Lhca1-9* genes are found to associate with the PSI core[12,13]. On the other hand, the major LHCII trimer is encoded by nine genes (*lhcbm1-9*) that have been divided into four groups based on their sequence homology: Type I (LHCBM3, LHCBM4, LHCBM6, LHCBM8, LHCBM9), Type II (LHCBM5), Type III (LHCBM2, LHCBM7), and Type IV (LHCBM1)[14]. In addition to the major LHCII trimers, only two minor antennae, CP26 and CP29, are found in green algal PSII-LHCII complex[15–18].

To maintain a maximal efficiency of photosynthetic electron transport and to avoid photodamage when exposed to excess light, plants and green algae redistribute the excitation energy between the two photosystems by moving part of LHCII through state transition[19–21]. It has been suggested that state transition is regulated by the redox state of the plastoquinone (PQ) pool[22]. In state 1, the PQ pool is oxidized and LHCII is bound to PSII, whereas in state 2, the PQ pool is reduced and the protein kinase, STN7 in higher plants or Stt7 in green algae, is activated through cytochrome $b_6f$ complexes[23,24]. The kinases phosphorylate the LHCII bound to PSII and a portion of the resultant phosphorylated LHCII dissociates from PSII and migrates to PSI, acting as a peripheral antenna for PSI[25–27]. Recent reports show that both Lhcb1 and Lhcb2 are heavily phosphorylated in higher plants in state 2, but only the phosphorylated Lhcb2 is present in the trimer bound to PSI[28–30]. This process is reversible, namely, the STN7/Stt7 kinase is inactivated upon reoxidation of the PQ pool, and a thylakoid-bound phosphatase dephosphorylates the LHCIIs and the dephosphorylated LHCIIs detaches from PSI and recouples to PSII, regenerating state 1[25–27].

In the past few years, PSI complexes associated with LHCIs and LHCIIs have been isolated from cells in state 2 from higher plants and green algae[30–36]. The structure of a PSI-LHCI-LHCII supercomplex from maize were analyzed by single-particle cryo-electron microscopy (cryo-EM) at 3.3 Å resolution[30], and two dimensional projection map of PSI-LHCI-LHCII supercomplex from *C. reinhardtii* has been obtained at 20 Å resolution[36]. These studies showed that while the plant PSI binds one LHCII trimer upon state transition, two LHCII trimers are moved to the green algal PSI upon state transition. However, the exact location and detailed association pattern of the LHCII trimers with the green algal PSI remain obscure due to the limited resolution. To understand the molecular details of this large supercomplex, we purified the PSI-LHCI-LHCII supercomplex from *C. reinhardtii* in state 2 and analyzed its structure using single-particle cryo-EM at an overall resolution of 3.42 Å.

Our structure revealed detailed organization and pigment arrangement of PSI-LHCI-LHCII from the green algae, enabling us to examine the excitation energy transfer (EET) pathways from the peripheral antennae to the core in a greater detail.

## Results

### Purification and characterization of the green algal PSI-LHCI-LHCII.
The PSI-LHCI-LHCII supercomplex was isolated from *C. reinhardtii* cells CC-137 in state 2 generated by the procedures described in Methods. At 77 K, fluorescence emission peaks originating from PSII and PSI were observed at 687 nm (F687) and 713 nm (F713), respectively. The state 1-induced cells showed a higher F687 value than F713, whereas the state 2-induced cells showed a higher F713 value than F687 (Fig. 1a). The changes of the spectral properties resulted from these treatments are consistent with the results from previous reports[34–36] and showed that the energy distribution between the two photosystems was altered and the cells were indeed locked in the state 1 and state 2 with less and more antennas bound to PSI, respectively. The thylakoid membranes isolated from the state 1 cells and state 2 cells also exhibited similar fluorescence emission spectra at 77 K as that of the two types of cells (Fig. 1b), indicating that the thylakoid membranes were still kept in state 1 or state 2 after purification.

Thylakoid membranes isolated from cells locked either in state 1 or state 2 were solubilized by n-dodecyl-α-D-maltoside (α-DDM) and loaded onto a sucrose density gradient. Green bands corresponding to PSI-LHCI complex were observed in both the state 1 and state 2 samples, whereas a new band corresponding to PSI-LHCI-LHCII complex was observed only in the state 2 sample (Fig. 1c). The PSI-LHCI-LHCII supercomplex was characterized by size-exclusion chromatography, electrophoresis, absorption and fluorescence spectroscopy (Figs. 1d–f and 2a). The results obtained show that the sample is homogenous and is composed of major components of PSI-LHCI and a large amount of antenna proteins from PSII. The protein bands corresponding to the extra LHCIIs found in the SDS-PAGE of PSI-LHCI-LHCII were analyzed by mass spectrometry, which showed that four types of LHCBM subunits, LHCBM3/4/8 (Type I), LHCBM5 (Type II), LHCBM2 (Type III), and LHCBM1 (Type IV) are present in the purified PSI-LHCI-LHCII supercomplex.

The phosphorylation state of individual proteins in the purified thylakoid membranes, PSI-LHCI and PSI-LHCI-LHCII supercomplex in state 2 were analyzed by either immunoblotting with an anti-phosphothreonine antibody (Fig. 2b) or Pro-Q Diamond Phosphoprotein Gel Stain that selectively stains phosphoproteins in the gel (Fig. 2c). The results indicate that mainly LHCBM3/4/8 (Type I) and LHCBM1 (Type IV) are phosphorylated, which is consistent with the results of a previous report[36]. HPLC analysis of PSI-LHCI and PSI-LHCI-LHCII supercomplexes purified from the state 2 cells indicates that the PSI-LHCI-LHCII supercomplex contains, in addition to Chl *a*, Chl *b*, violaxanthin (Vio) and lutein (Lut) found in the PSI-LHCI complex, much more LHCII pigments including neoxanthin (Neo), loroxanthin (Lor) and a larger amount of β-carotenes (BCRs) compared to that of the PSI-LHCI supercomplex (Supplementary Fig. 1). The HPLC measurements also show the presence of a small amount of a pigment involved in the xanthophyll cycle, antheraxanthin (Ant), in both the PSI-LHCI and PSI-LHCI-LHCII supercomplexes isolated from the state 2 cells. Since this pigment was not found in PSI-LHCI isolated from state 1 cells of *C. reinhardtii*[12], this indicates that the xanthophyll cycle may occur during the state transition of *C. reinhardtii*[37].

### Overall structure.
The structure of the PSI-LHCI-LHCII supercomplex from *C. reinhardtii* (Cr) was determined using single-particle cryo-EM analysis with the homogenous sample prepared

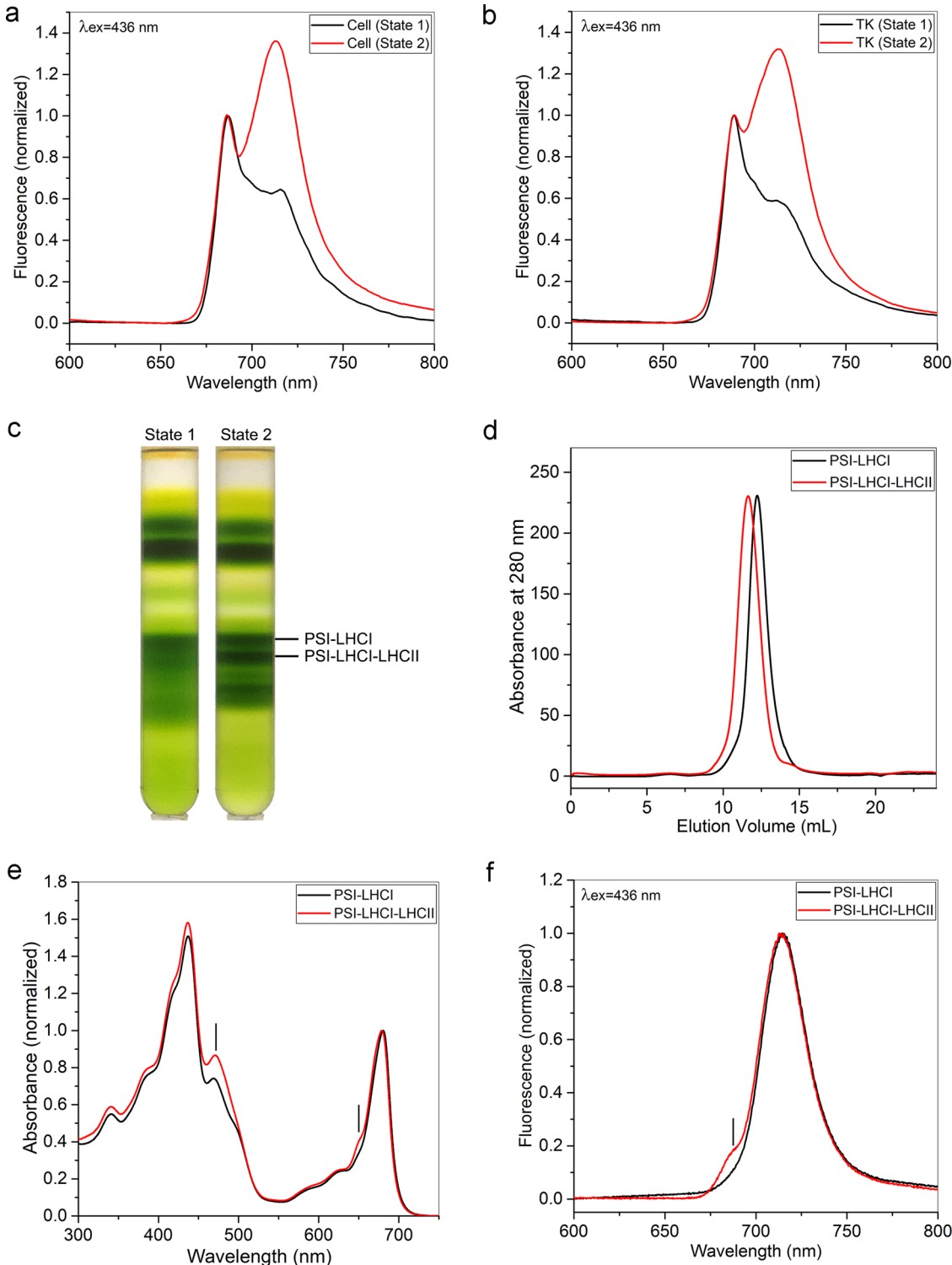

**Fig. 1 Sample preparation and characterization of the PSI-LHCI-LHCII supercomplex from *C. reinhardtii*. a**, **b** Low temperature (77 K) fluorescence emission spectra of *C. reinhardtii* cell (**a**) and thylakoid (**b**) locked in state 1 (black line) and state 2 (red line). The excitation wavelength was 436 nm and all spectra were normalized to the emission value at 688 nm. **c** Separation of PSI-LHCI-LHCII supercomplex by sucrose density gradient (SDG) centrifugation from cells in state 1 and state 2. **d** Size-exclusion chromatographic elution profiles of the PSI-LHCI and PSI-LHCI-LHCII fractions isolated by SDG from the state 2 cell. Elution was performed with a Superose 6 Increase 10/300 GL column (flow rate of 100 µl min⁻¹) at 4 °C and monitored by absorption at 280 nm. **e** Room-temperature absorption spectra of the PSI-LHCI and PSI-LHCI-LHCII obtained from size-exclusion chromatography. The spectra were normalized to the maximum in the red region. The PSI-LHCI-LHCII supercomplex showed higher peaks at 470 nm and 650 nm (indicated by bars), indicating that this fraction contains higher Chl *b* content (from LHCII) than the PSI-LHCI complex. **f** Low temperature (77 K) fluorescence emission spectra of PSI-LHCI and PSI-LHCI-LHCII after size-exclusion chromatography upon excitation of Chl *a* at 436 nm. The spectra are normalized to the maximum of the emission peaks. Both two samples show the typical emission peaks of PSI-LHCI in red region, but the PSI-LHCI-LHCII has a small shoulder at 680 nm (indicated by bar).

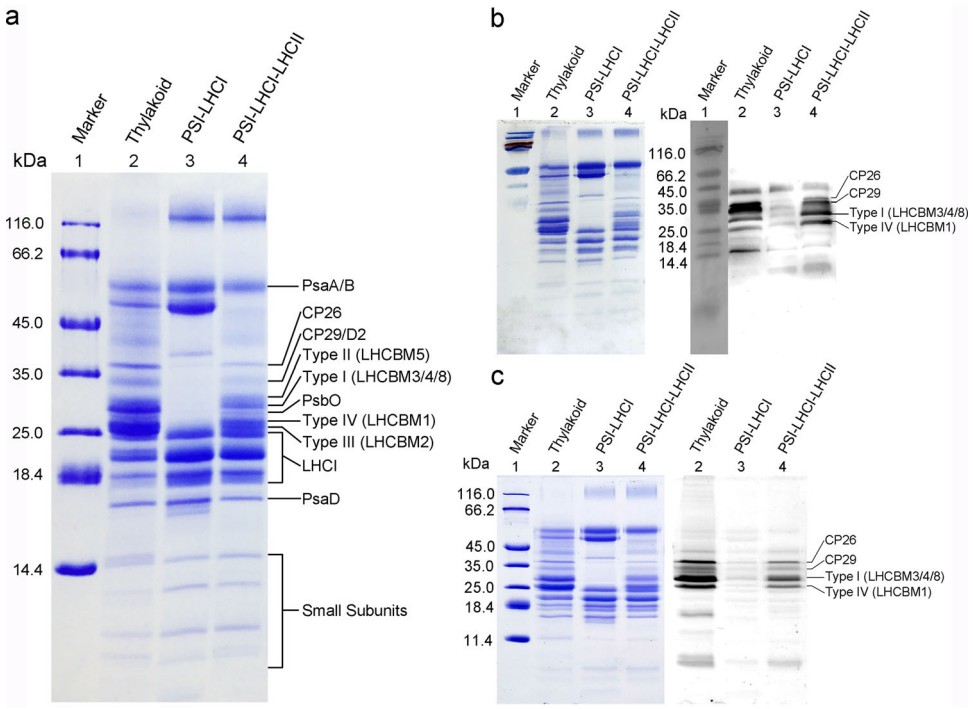

**Fig. 2 SDS-PAGE analysis and identification of phosphorylation state of proteins in the PSI-LHCI-LHCII supercomplex purified from *C. reinhardtii* cells in state 2. a** SDS-PAGE analysis of the PSI-LHCI and PSI-LHCI-LHCII purified from *C. reinhardtii* cells locked in state 2. The proteins of the Coomassie bands was identified based on the mass spectrometry analysis. In the band of Type I LHCBM, the amount of LHCBM4 is relatively lower than that of LHCBM3, but higher than that of LHCBM8. **b** Immunoblot analysis of the phosphoproteins in PSI-LHCI-LHCII supercomplex using a phosphothreonine antibody. The polypeptide samples were separated by SDS-PAGE and stained with Coomassie blue (left), and polypeptides in the gels were electrophoretically transferred to a nitrocellulose membrane and proteins were detected with a PThr antibody (right). **c** Detection of the phosphoproteins by Pro-Q Diamond staining. The polypeptide samples were separated by SDS-PAGE and stained with Coomassie blue (left), and then the same gel was stained with Pro-Q Diamond to detect the presence of phosphoproteins (right). Lane 1: marker; lane 2: thylakoid membranes; lane 3: PSI-LHCI after size-exclusion chromatography; lane 4: PSI-LHCI-LHCII after size-exclusion chromatography. Samples were loaded onto SDS-PAGE at 20, 5, and 5 μg of Chl per lane in **a**, **b**, and **c**, respectively. The SDS-PAGE and immunoblot analysis were conducted more than three times with samples independently purified, and all results are similar to the ones shown here.

by size-exclusion chromatography. We collected 5087 cryo-EM micrographs, and picked 1,582,470 particles for subsequent data processing. After 2D and 3D classifications (Supplementary Fig. 2 and Supplementary Table 1), particles with a major population were selected and processed, which yielded a cryo-EM map with an overall resolution of 3.42 Å for the whole PSI-LHCI-LHCII supercomplex.

The overall structure of this Cr-PSI-LHCI-LHCII is composed of two moieties: the PSI-LHCI supercomplex and two LHCII trimers (Fig. 3a, b). The moiety of PSI-LHCI is similar to the Cr-PSI-10 Lhcas reported previously[12,13], except for the presence of PsaO, an important PSI core subunit. This subunit was absent in the crystal structure of higher plant PSI-LHCI[6,7] and cryo-EM structures of green algal PSI-LHCI[12,13,38], but found in the cryo-EM structures of red algal PSI-LHCI[39] and higher plant PSI-LHCI-LHCII[30]. Two LHCII trimers are associated with PSI-LHCI; they are located close to each other at the PsaO-PsaL-PsaH-Lhca2 side. One LHCII trimer (LHCII-1) attaches to the PSI core by PsaO, PsaH, and PsaL subunits, whereas the other LHCII trimer (LHCII-2) attaches to the Lhca2 and LHCII-1 subunits. The local resolution of LHCIIs is lower than that of PSI-LHCI, suggesting that these two LHCIIs have a higher mobility and hence are loosely bound and easy to be detached from PSI-LHCI. However, we were able to resolve the amino acid residues in the interfaces between LHCII-1 and the PSI core (see below).

Superposition of the Cr-PSI-LHCI-LHCII structure with that of a higher plant PSI-LHCI-LHCII based on the PSI core revealed important differences in the location of the LHCII trimers.

The LHCII-1 trimer in Cr-PSI-LHCI-LHCII is more adjacent to PsaH/PsaL rather than to PsaK, and moves around 19 Å toward PsaH along with a clockwise rotation by 40° compared with the position of LHCII in maize PSI-LHCI-LHCII. In addition, the bound LHCII moiety is tilted slightly to the lumenal side instead of the stromal side (Fig. 3c, d) in the Cr-PSI-LHCI-LHCII, indicating a species dependent organization of the supercomplex.

In addition to the protein subunits, we identified 332 Chls, 83 carotenoids, 3 $Fe_4S_4$ clusters and some lipids in the PSI-LHCI-LHCII supercomplex (Supplementary Fig. 3, 4 and Supplementary Table 2). Compared with the pigments in the previous Cr-PSI-LHCI structures, some new pigments around PsaH/PsaL/PsaO were observed in the Cr-PSI-LHCI-LHCII (Supplementary Fig. 4a), which may be required for the association of the additional LHCII subunits and/or EETs (see below). Due to the limited resolution, it was not possible to distinguish between Chl *a* and Chl *b* in most cases. Thus, we modeled these pigments according to the high-resolution structures of Cr-PSI-LHCI[13] and isolated LHCII[8] reported previously. In total, 33 lipids (consisted of four types of lipid molecules) were identified in the Cr-PSI-LHCI-LHCII, among which, 27 are located in the PSI-LHCI moiety and 6 are located in the two LHCII trimers (Supplementary Fig. 4b).

**Structure of the PSI-LHCI moiety.** Structural comparison of the Cr-PSI-LHCI-LHCII with other reported green algal PSI-LHCI[12,13,38], red alga PSI-LHCI[39], and higher plant

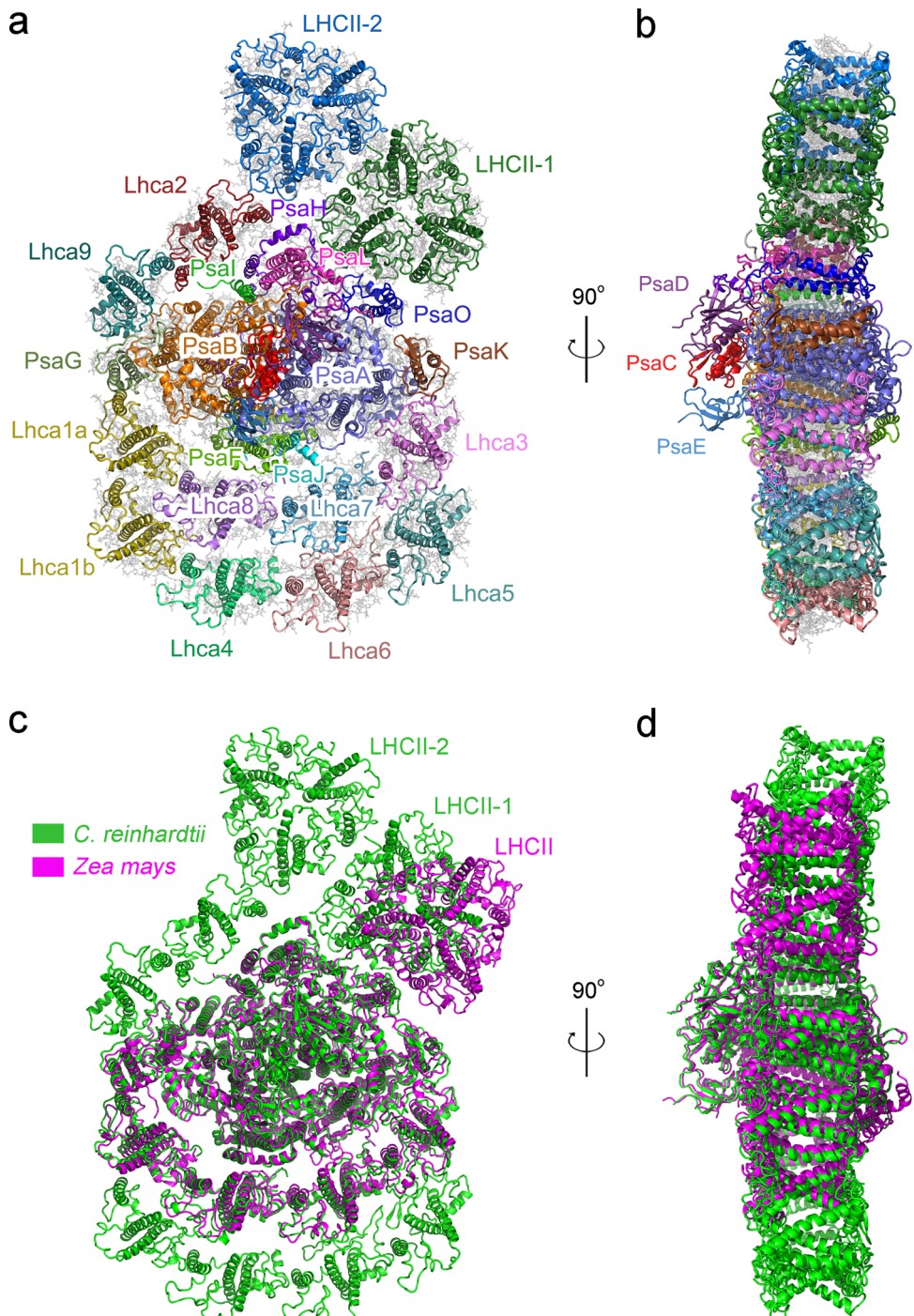

**Fig. 3 Over architecture of the PSI-LHCI-LHCII supercomplex of *C. reinhardtii* and its comparison with that of maize. a**, **b** Overall structure the Cr-PSI-LHCI-LHCII supercomplex viewed along the membrane normal from the stromal side (**a**) and along the membrane plane (**b**). Protein subunits in **a** and **b** are shown in cartoon model and colored differently. **c**, **d** Comparison of the overall structures of Cr-PSI-LHCI-LHCII (green) and higher plant maize PSI-LHCI-LHCII (PDB code 5ZJI, magenta), viewed along the membrane normal from the stromal side (**c**) and along the membrane plane (**d**).

PSI-LHCI-LHCII[30] supercomplexes reveals some similarities as well as differences in the structure of PSI-LHCI (Figs. 3c, d, and 4). The PSI-LHCI moiety of Cr-PSI-LHCI-LHCII binds 10 Lhca subunit, which is the same as the previously reported green algal PSI-LHCI structures (Fig. 4a–d and Supplementary Figs. 5–7)[12,13,38]. On the basis of the cryo-EM density, the complete structures of three core subunits PsaH, PsaL, and PsaO were identified in the current Cr-PSI-LHCI-LHCII, whereas these subunits were absent or not well identified in the previously

solved PSI-LHCI structures of the green algae[12,13,38]. These subunits are located in the interface between PSI-LHCI and LHCII, and provide binding sites for LHCII during the state transition. The present results suggest that binding of these subunits to PSI-LHCI is enhanced by the binding of LHCII, which prevented these subunits from releasing during purification. Additional pigments (Chls and BCRs) were found to bind mainly to the PsaH, PsaL and PsaO subunits in the interface between the PSI core and LHCII (Fig. 4e–g). Most of these

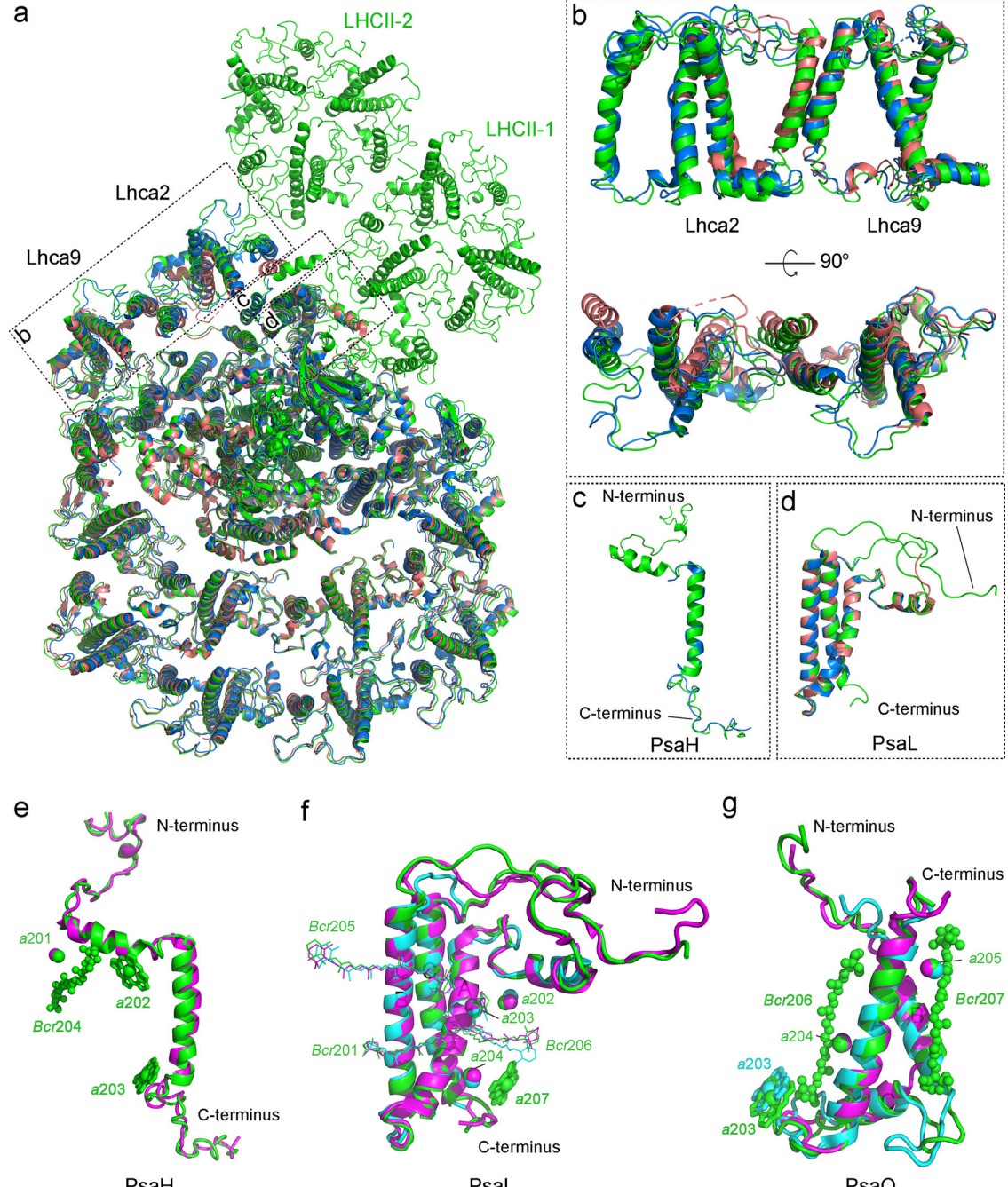

**Fig. 4 Structures of the PSI-LHCI moiety and individual subunits in the PSI-LHCI-LHCII supercomplex of *C. reinhardtii* and their comparisons with those from the PSI-LHCI of red alga, green alga and the PSI-LHCI-LHCII of higher plant. a** Structural superposition of Cr-PSI-LHCI-LHCII (green) and two Cr-PSI-LHCI supercomplexes (PDB code 6JO5 and 6IJO, deep-salmon and marine) viewed from the stromal side. **b–d** Enlarged views of the Lhca2/Lhca9 subunits (**b**), PsaH subunits (**c**), and PsaL subunits (**d**) in **a**. **e** Structural comparison of the PsaH subunits from the PSI-LHCI-LHCII of green alga *C. reinhardtii* (green) and higher plant maize (PDB code 5ZJI, magenta). **f**, **g** Structural comparison of the PsaL subunits (**f**) and PsaO subunits (**g**) from the green alga *C. reinhardtii* (green), the red alga *C. merolae* (PDB code 5ZGB, cyan) and higher plant maize (PDB code 5ZJI, magenta). The newly discovered chlorophylls and carotenoids in **e**, **f**, and **g** are shown in bold sticks and spheres, respectively.

pigments are involved in EETs from LHCII to the PSI core. Meanwhile, the PsaN subunit, which has been observed only in the maize PSI-LHCI-LHCII structure so far[30], is absent in our structure.

The structure of PsaH subunit is similar to that of higher plant PSI-LHCI-LHCII, containing a transmembrane (TM) helix and a helix parallel to the membrane plane in the stromal side (Fig. 4e). The N-terminal loop-helix-loop region, which extensively interacts with the PsaL and PsaD subunits in the stromal side, was

observed in our structure but not in the structures of previously reported Cr-PSI-LHCI[12,13]. Three Chls (Chl *a*201, Chl *a*202, and Chl *a*203) and one BCR (Bcr204) were found to bind to PsaH in the present structure, whereas no pigment binding sites were identified in the PsaH structure of green algae (*C. reinhardtii*, *Bryopsis corticulans*)[12,13,38] (Supplementary Figs. 5c and 7c) and only one Chl (Chl *a*201) was found in PsaH of the maize PSI-LHCI-LHCII[30] (Fig. 4e). The newly found Chl *a*202 and Chl *a*203 in PsaH may mediate EET from Lhca2 to the PSI core.

The structure of the green algal PsaL shares a high similarity with that of higher plant PsaL[30] and has longer N and C-termini as well as a larger loop between the second and third TM helices than that of red algal PsaL (Fig. 4f)[39]. In the stromal side, the two loop regions have close interactions with the phosphorylation tail of LHCII and the N-terminal tail of PsaO, respectively. Four Chls (Chl a202, Chl a203, Chl a204, and Chl a207) and three BCRs (Bcr201, Bcr205 and Bcr206) were observed in the green algal PsaL. Among the four Chls, Chl a207 was absent in PsaL of previously reported structures of red alga[39], green algae[12,13,38], and maize[30], and is located in the C-terminal region of the lumenal side adjacent to Chl a204, forming a Chl pair to participate in EET from LHCII trimer-1 to the PSI core (Fig. 4f, Supplementary Figs. 5d, 6d, and 7d).

PsaO is involved in the connection between LHCII and the PSI core as that in the higher plant PSI-LHCI-LHCII[30]. The structure of the green algal PsaO contains two TM helixes and one amphipathic helix at the lumenal side, resembling that of higher plant[30] and red alga[39], but some differences were found in the N-terminal and loop regions. In the stromal side, the Cr-PsaO has a similar N-terminal loop motif as that of higher plant and this loop participates in the interactions with the LHCII phosphorylation site to stabilize LHCII binding. The N-terminal tails of higher plant and green algal PsaO are longer than that of red alga (Fig. 4g); this extra N-terminal region facilitates its interaction with the PsaD subunit. In the lumenal side, the loop region between helix B and helix C in Cr-PsaO is similar to that of red alga and mediates its association with PsaA. In addition, the Cr-PsaO binds three Chls (Chl a203, Chl a204, Chl a205) and two BCRs (Bcr206 and Bcr207), whereas only two Chls (Chl a204, Chl a205) were found in PsaO of higher plant and no BCR was found in PsaO of red alga and higher plant. The new Chl a203 found in the Cr-PsaO facilitates EET from LHCII to PsaA. The orientations of the two BCRs (Bcr206 and Bcr207) in the Cr-PsaO are parallel to the chlorin plane of one or more of the three Chls, which may be beneficial for the function of photoprotection.

In addition to the above differences, the stromal loop regions of PsaG and PsaK were identified in the structure Cr-PSI-LHCI-LHCII (Supplementary Fig. 6). The loop region of PsaG is located between Lhca9 and PsaB, enhancing the binding of these two subunits, whereas the loop region of PsaK mainly interacts with PsaA, resulting in the stable binding of PsaK in the PSI core.

The LHCIs in the side layer (Lhca2 and Lhca9) have been shown to have a low occupancy and are loosely associated with the core in the previously reported Cr-PSI-LHCI structures[12,13], but showed a high-quality cryo-EM density in our PSI-LHCI-LHCII sample (Supplementary Fig. 8a). Detailed structures of Lhca2 and Lhca9 were identified, indicating that they are bound to the PSI core by PsaH and PsaG, respectively. Lhca2 has a unique structure consisted of four transmembrane helices (named A, B, C, F from N- to C-terminus) and both the N-terminus and C-terminus are located in the stromal side, which closely resemble that from the green alga *Bryopsis corticulans*[38]. Helix F of Lhca2 runs parallel to helix C of Lhca9, and these two helices interact extensively through hydrophobic interactions to connect them (Supplementary Fig. 8b). Two Chls a (Chl a601/Chl a614) located in the helix F side of Lhca2 interact with helix C of Lhca9, and two Chls (Chl a606/Chl a609) of Lhca9 interact with helix F of Lhca2, which strengthen the connections between Lhca2 and Lhca9. Helix B of Lhca2 interacts with the transmembrane helix of PsaH through a monogalactosyldiacylglycerol molecule, whereas Chl a202 from PsaH interacts with Chl a609 of Lhca2, which are beneficial for the stable association of Lhca with the PsaH side (Supplementary Fig. 8c). Our structure also shows more lipids at the interface between LHCIs and the PSI core,

which may play important roles in the assembly of the PSI core and peripheral antenna complexes (Supplementary Fig. 4b).

**Assembly of LHCIIs in the PSI-LHCI-LHCII supercomplex.** The most significant feature of Cr-PSI-LHCI-LHCII is the binding of two additional major LHCII trimers (LHCII-1 and LHCII-2), which increases the EETs from LHCII to PSI during the process of state transition. LHCII-1 is the same as that found in the maize PSI-LHCI-LHCII, but its position was shifted to the LHCII-2/Lhca2 side significantly (Fig. 3c). In the binding of LHCII-1 to the PSI core, one monomer (monomer-1) of the LHCII-1 interacts with the core subunits of PsaH/PsaL/PsaO at the stromal side. The phosphorylation of LHCII in state 2 of *C. reinhardtii* was detected by Pro-Q Diamond stained SDS-PAGE and immunoblot analysis using a phosphothreonine antibody (Fig. 2b, c). Both Pro-Q Diamond and immunoblot analysis showed phosphorylation in both LHCBM1 (Type IV) and LHCBM3/4/8 (Type I), and the immunoblot analysis showed phosphorylation of the threonine residue in these LHCBMs. The high-quality cryo-EM density allowed construction of the complete N-terminal region and the phosphorylated threonine residue (pThr) of monomer-1 of LHCII-1 (Supplementary Fig. 9). The cryo-EM density around the N-terminal tail of monomer-1 of LHCII-1 matched with the conserved sequence "RRTV" of LHCBM1 as well as "KKTA" of LHCBM4/8, but not with the N-terminal "GKTA" sequence of LHCBM3, where the residue Thr is phosphorylated. Based on the higher abundance of LHCBM4 compared to LHCBM8, and the fact that deletion of LHCBM1 does not inhibit the state transition of *C. reinhardtii* as reported previously[40,41], we tentatively assign phosphorylated LHCBM4 to the LHCII-1 subunit involved in the interactions with the PSI core subunits. The cryo-EM density of the N-terminal region can also be fitted with the N-terminal sequence of LHCBM1 (RRTV). If this is the case, there is a possibility that deletion of LHCBM1 may cause other LHCBMs to be phosphorylated and moved to PSI under state transition conditions. The phosphate group of pThr strongly interacts with several residues from the loop region between helix B and helix C of PsaL (Figs. 5 and 6). The two basic residues (Lys1 and Lys2) ahead of pThr interact with PSI core subunits and contribute to the binding of LHCII-1. Lys1 interacts with the N-terminal regions of PsaH and PsaL, whereas Lys2 interacts with residues from the N-terminal loop of PsaO and PsaL (Fig. 5). It is interesting to note that the binding site of pThr is evolutionarily and structurally conserved in both green alga *C. reinhardtii* and the high plant maize PSI-LHCI-LHCII (Supplementary Fig. 10). In addition, the AC loop of LHCII-1 (monomer-1) interacts with the stromal loop and Chl a201 of PsaH, and the BC loop in the lumenal side of LHCII-1 (monomer-1) interacts with the C-terminal region and Chl a207 of PsaL (Fig. 6a, b). Chl a202 of PsaL also interacts with the loop region between helices B and C of LHCII-1 at the stromal surface to help its binding (Fig. 6b).

PsaO has been reported to be involved in the binding of LHCII to PSI during the state transition in higher plants[30,42]. A similar function was observed for PsaO of *C. reinhardtii* (Fig. 6c). In the stromal side, the C-terminal region of PsaO interacts with the loop region of PsaA, and Bcr850 of PsaA interacts with helix A of PsaO. In the lumenal side, both the AC loop and BC loop of PsaO interact with the helix of PsaA, while the helix C of PsaO has a close connection with helix D of LHCII-1 (monomer-1). In addition, the Chls and carotenoids of PsaO are involved in the interactions with the pigments from the adjacent LHCII-1 and PsaA, and mediate the EET from LHCII-1 to the PSI core. Thus, LHCII-1 (monomer-1) and PsaH/PsaL/PsaO form a stable

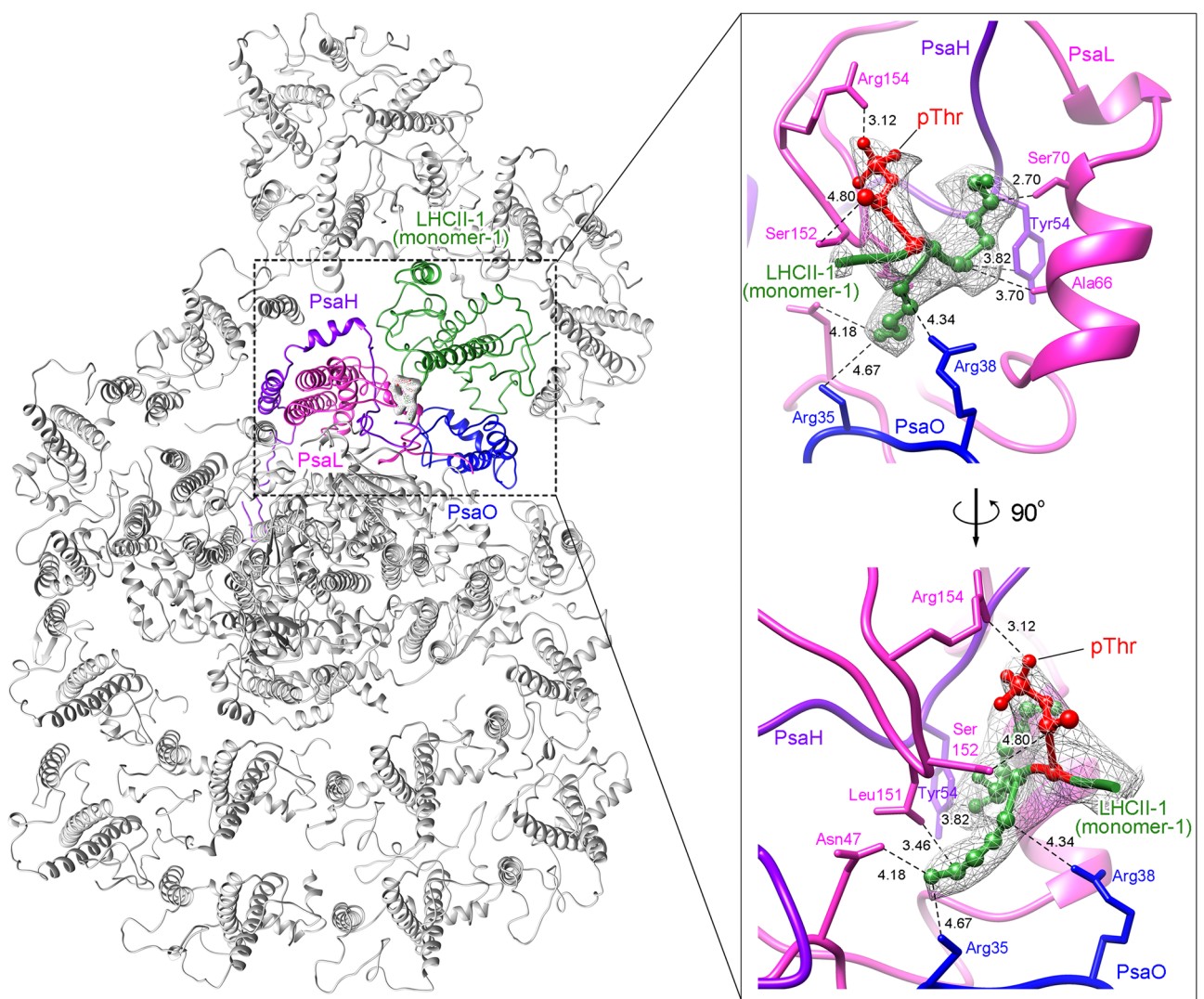

**Fig. 5 The structure of the phosphorylation site of LHCII and its interactions with the PSI core subunits of _C. reinhardtii_.** The LHCII residues involved in the interactions with PSI core subunits are shown in ball and stick, and the densities of these residues are shown as gray meshes, in the right-side panel. The residues of PSI core subunits involved in the interactions with the phosphorylated LHCII are shown as sticks. The dashed lines indicate hydrogen bonds or van der Waals interactions between adjacent groups. The distances (Å) between two connected atoms are labelled nearby the dashed line.

triangle architecture, which plays a vital role for the binding of LHCII-1 to the PSI core and EET in the state transition.

LHCII-2 is found in the Cr-PSI-LHCI-LHCII only but not in the maize PSI-LHCI-LHCII, and its binding increases cross-section of light harvesting in the Cr-PSI-LHCI-LHCII. LHCII-2 is bound to the PsaH/Lhca2 side adjacent to LHCII-1; however, it shows a low occupancy probably due to its flexibility and weak binding to the PSI core. LHCII-2 has no direct interactions with the PSI core subunits, but associates with the PSI core through Lhca2 and the LHCII-1 trimer.

**Possible excitation energy transfer pathways**. The EET pathways in Cr-PSI-LHCI-LHCII are partly similar to those of Cr-PSI-LHCI or maize PSI-LHCI-LHCII owing to their structural similarity. However, some distinct features such as the LHCII orientation and appearance of new pigments result in some EET pathways different from those of plant PSI-LHCI-LHCII or Cr-PSI-LHCI. Based on the distribution of pigments and the distances between Chls, we identify the plausible EET pathways in the stromal and lumenal sides of Cr-PSI-LHCI-LHCII (Fig. 7a, b).

In the stromal layer, the edge-to-edge distances between Chls of Lhca8/Lhca7 and PSI core are around 23.0 Å or longer; thus direct EET from these Lhcas to the PSI core is rather inefficient. EET from the outer belt LHCIs to PSI core mainly occurs through two pathways involving Lhca1b-Lhca4-Lhca8-Lhca1a-PsaB and Lhca6-Lhca5-Lhca7-Lhca3-PsaA (Fig. 7a). In the first case, energy absorbed by the outer antenna Lhca1b may be transferred to PsaB through Lhca1a, whereas energy absorbed by Lhca4 may be transferred to PsaB through Lhca8-Lhca1a. In the second case, energy absorbed by the outer antenna Lhca6 may be delivered to PsaA through Lhca5, Lhca7, and Lhca3. The energy may also be transferred between Lhca4 and Lhca6 at the outer belt and between Lhca8 and Lhca7 at the inner belt, enabling energy equilibrium between the two pathways. In addition, the Lhca2/Lhca9 heterodimer located at the PsaH/PsaI/PsaB/PsaG side may transfer their absorbed energy to the PSI core via PsaB/PsaH/PsaG. Indeed, the edge-to-edge distance of Lhca2-Chl _a_609 and PsaH-Chl _a_202 is 3.8 Å, providing an efficient EET pathway (Fig. 7c).

The excitation energy absorbed by the two LHCII trimers may be transferred to the PSI core through either direct or indirect routes. In the stromal side, the closest edge-to-edge distance between the

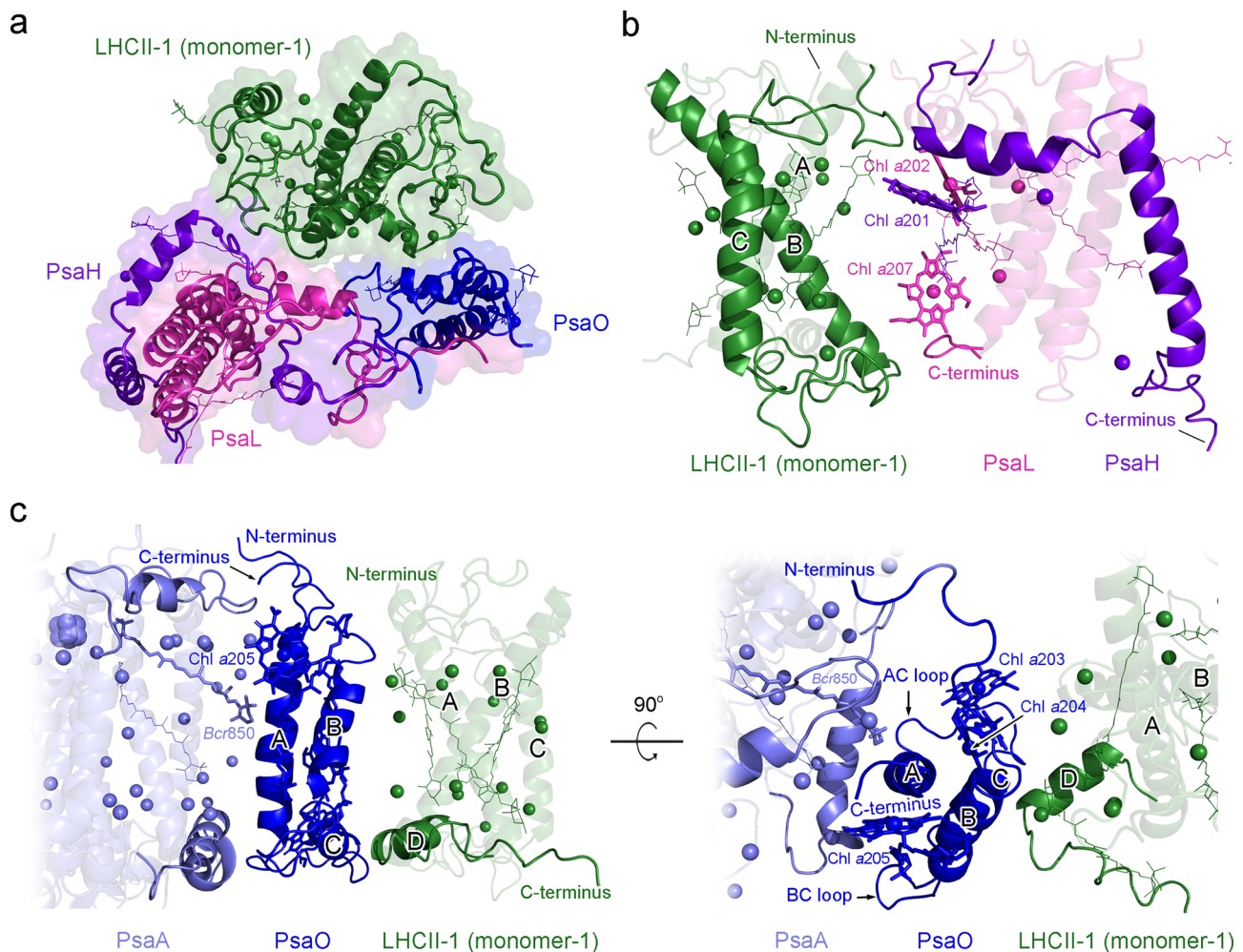

**Fig. 6 Interactions between the N-terminal tail of LHCII-1 (monomer-1) with the PSI core subunits in the PSI-LHCI-LHCII supercomplex of *C. reinhardtii*. a** Surface and cartoon representations of the local structure including LHCII-1 (monomer-1), PsaH, PsaL, and PsaO, viewed from the stromal side. **b** Interactions between LHCII-1 (monomer-1) and PsaL as well as PsaH. **c** Interactions between LHCII-1 (monomer-1) and PsaA mediated by PsaO.

Chls in LHCII-2 and PSI core is around 18 Å, which exists between Chl $a612$ of LHCII-2 and Chl $a202$ of PsaH. This distance may allow EET from LHCII-2 to the PSI core, but may not be efficient. On the other hand, LHCII-2 is close to LHCII-1, and the EET from LHCII-2 may be mediated by LHCII-1. Some new pigments found in the interface between LHCII-1 and PSI core in the stromal side may mediate this EET. The edge-to-edge distance between Chl $a610_{LHCII-1\ (monomer-1)}$ to Chl $a201_{PsaH}$ and Chl $a202_{PsaL}$ is 13.6 Å and 14.9 Å (Fig. 7d). Chl $a201_{PsaH}$, Chl $a202_{PsaL}$ and Chl $a610_{LHCII-1\ (monomer-1)}$ form a pigment cluster, and thus play a vital role in mediating EET from LHCII to PSI core in the stromal side.

In the lumenal side, the EET pathways from LHCI or LHCII to the PSI core seem to be very efficient (Fig. 7b). The energy harvested by Lhca1b may be transferred to Lhca1a, and further to PsaB through PsaF. Alternatively, energy harvested by Lhca1b and Lhca4 may be transferred to PsaA through Lhca8 and PsaJ. Lhca3 is located in a crucial position mediating the EET from LHCIs to the PSI core. Energy harvested by Lhca6 may be transferred to Lhca3 by Lhca7, whereas energy harvested by Lhca5 may directly transfer to Lhca3. As a consequence, Chl $a607$ in Lhca3 transfers energy collected by Lhca3 to Chl $a817$ of PsaA. Lhca2/Lhca9 heterodimer may transfer the excitation energy to PsaH and PsaB in the lumenal side. The edge-to-edge distance between Chl $a607_{Lhca2}$ and Chl $a203_{PsaH}$ is 10.2 Å (Fig. 7c), thus EET may occur between this Chl pair.

The Chls in LHCII-2 are close to both Lhca2 and LHCII-1 in the lumenal side, but far from any of the PSI core subunits; thus, LHCII-2 may transfer energy to the PSI core through Lhca2 and LHCII-1 (Fig. 7e). The edge-to-edge distances between Chl $b605_{LHCII-2}$ and Chl $a604_{Lhca2}$, and between Chl $b605_{LHCII-2}$ and Chl $a606_{Lhca2}$ are 10.7 Å and 10.2 Å, respectively, enabling EET from Chl $b605_{LHCII-2}$ to Chl $a604_{Lhca2}$ and Chl $a606_{Lhca2}$. The edge-to-edge distances between Chl $a614_{LHCII-2}$ to Chl $b605_{LHCII-1\ (monomer-1)}$ and Chl $a614_{LHCII-1\ (monomer-2)}$ are 6.0-6.1 Å, indicating their efficient EET.

The Chls in LHCII-1 may transfer the excitation energy to the PSI core through PsaL and PsaO subunits in the lumenal side. One of the main EET routes from LHCII-1 to the PSI core may occur through the pathway involving Chl $a604_{LHCII-1\ (monomer-1)}$ and Chl $a207_{PsaL}$. The plane of Chl $a207_{PsaL}$ is parallel to that of Chl $a204_{PsaL}$ with a plane-to-plane distance of around 3 Å, and thus these two Chls are strongly coupled to form a low-energy Chl dimer (Fig. 7b). Chl $a604_{LHCII-1\ (monomer-1)}$ may transfer the excitation energy to the Chl pair $a207$-$a204$ in PsaL, and further to Chl $a835$ in PsaB.

The edge-to-edge distance between Chl $a614_{LHCII-1\ (monomer-1)}$ and Chl $a204_{PsaO}$ is 6.2 Å (Fig. 7f), enabling EET from Chl $a614$ of LHCII-1 (monomer-1) to Chl $a204$ of PsaO. The Chl $a204_{PsaO}$ may subsequently transfer the excitation energy to Chl $a203_{PsaO}$ and further to the Chl pair $a204$-$a207$ in PsaL. Chl $a204_{PsaO}$ may also transfer the excitation energy to Chl $a205_{PsaO}$ and Chl

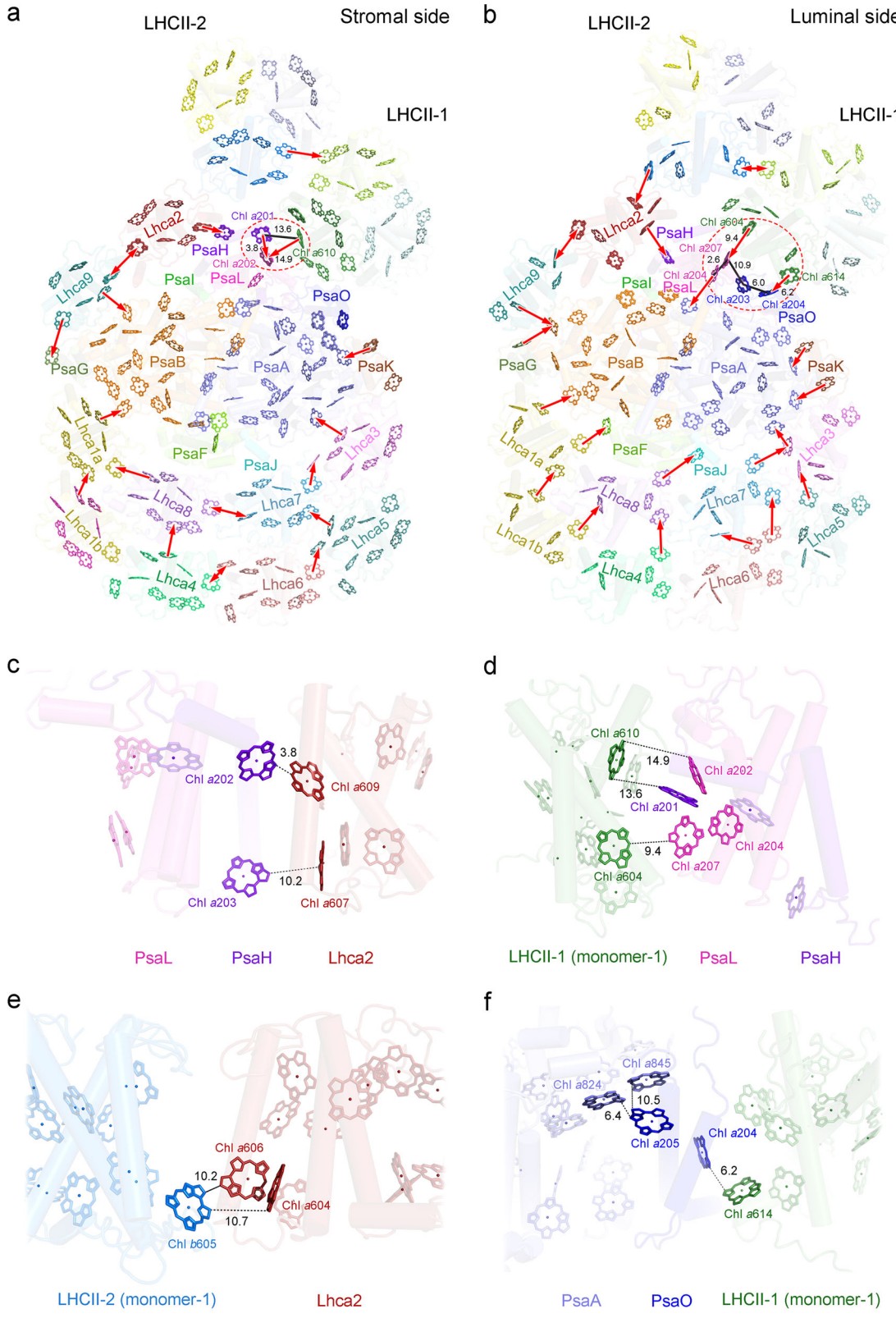

**c** PsaL PsaH Lhca2

**d** LHCII-1 (monomer-1) PsaL PsaH

**e** LHCII-2 (monomer-1) Lhca2

**f** PsaA PsaO LHCII-1 (monomer-1)

$a845_{PsaA}$ in the stromal side, with edge-to-edge distances of 15–16 Å. All these Chl networks enable efficient EET in the supercomplex.

Owing to the binding of LHCII-1 to the PSI core, several new pigments in the interfacial region between the PSI core and LHCII exhibited a high-quality cryo-EM density and were thus identified, which is in accordance with the results from HPLC measurements (Supplementary Fig. 1). In the PsaL, the newly found Chl $a207$ is closely coupled with Chl $a204$ to form a low-energy Chl pair, and this Chl pair forms a pigment cluster with the nearby $Bcr206_{PsaL}$ and $Bcr204_{PsaH}$ (Fig. 8a). In the PsaO subunit, two BCR binding sites were found and one BCR ($Bcr206$) is close to the Chl $a203$ and Chl $a204$, which may contribute to either the efficient EET from PsaO to PSI core or the quenching

**Fig. 7 Pigment arrangement and possible excitation energy transfer pathways from peripheral antenna complexes to the reaction center in the PSI-LHCI-LHCII supercomplex of *C. reinhardtii*. a, b** Chlorophyll distribution and possible EET pathways within the whole PSI-LHCI-LHCII supercomplex at the stromal side (**a**) and lumenal side (**b**). All protein subunits are shown in transparent cartoon model, and the Chls are shown in different colors as that of protein subunits shown in Fig. 3a. For clarity, the phytol tails of the Chls are omitted. Red arrows indicate possible EET pathways from the peripheral antennae to PSI core and among the antennae. The dashed elliptical rings indicate the regions in which pigment clusters are formed. **c–f** The interfacial chlorophylls between Lhca2 and PsaH or PsaL (**c**), LHCII-1 (monomer-1) and PsaL or PsaH (**d**), LHCII-2 (monomer-1) and Lhca2 (**e**), PsaO and PsaA or LHCII-1 (monomer-1) (**f**). Chlorophylls involved in the possible EETs are highlighted as sticks and labeled with the same color as that of the protein subunits. The edge-to-edge distances (Å) between two adjacent interfacial chlorophylls are indicated in black.

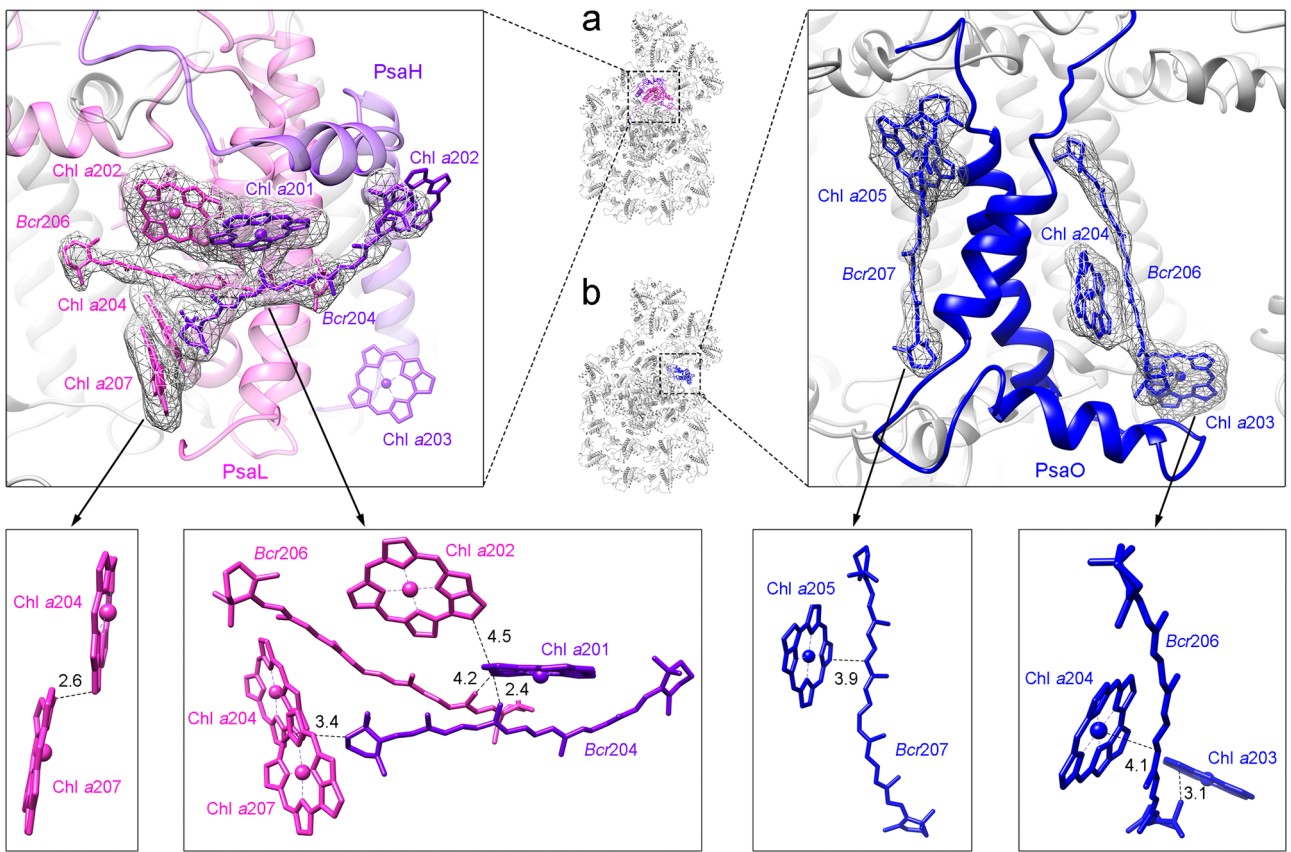

**Fig. 8 Distribution pattern of typical chlorophylls and β-carotenes involved in excitation energy transfer from LHCII to the PSI core or photoprotection triggered by highlight. a, b** Typical chlorophylls and β-carotenes around PsaH/PsaL (**a**) and PsaO (**b**). Chlorophylls and β-carotenes involved in the possible EETs or photoprotection are highlighted as sticks and labeled with the same color as that of the protein subunits. The nearest distances (Å) between two adjacent pigments are indicated in black.

of Chl triplet states under highlight conditions (Fig. 8b). The short distance between these pigments and their specific locations suggest that this cluster is an important site for photoprotection.

## Discussion

The Cr-PSI-LHCI-LHCII structure solved in this study provides the first near-atomic structure of the PSI-LHCI-LHCII super-complex involved in the state transition of green algae. The Cr-PSI-LHCI-LHCII binds two LHCII trimers, which is one more than the higher plant maize PSI-LHCI-LHCII. This probably reflects the different light environment that different organisms experience. Higher plants live on land where a relatively strong light intensity is available. Thus, higher plant PSI binds a relatively small number of LHCI (four), and also a smaller number of LHCII upon state transition. On the other hand, green algae grow under water or in soil where light intensity is limited; thus, their PSI usually binds much more number of LHCI (ten), and also requires an extra LHCII trimer than that of higher plant PSI under the state 2 condition.

In the PSI-LHCI moiety, the well-defined cryo-EM map allowed us to identify the complete structure of PSI core subunits, PsaH and PsaL, as well as LHCIs subunits Lhca2 and Lhca9, which were not well modeled in the previous studies. In addition, another PSI core subunit PsaO observed in the structures of red algal PSI-LHCI and maize PSI-LHCI-LHCII but not in green algal PSI previously, was found in the present structure. Previous functional and structural studies have shown that PsaO serves as the anchoring site for LHCII in state transition of higher plants[30], and deletion of this subunit led to the decrease of the degree of state transition in *Arabidopsis*[42]. The present Cr-PSI-LHCI-LHCII structure shows that PsaO mediates the direct connection of LHCII with the PSI core and also mediates EET from LHCII to the PSI core, in agreement with its importance for the formation of the PSI-LHCI-LHCII supercomplex. This suggests that PsaO is involved in the state transition to cope with the highly fluctuating light conditions.

The green algal PSI-LHCI-LHCII shows that two LHCII trimers are mobile in *C. reinhardtii*, although it was reported that

CP29 may also be involved in the migration to PSI during the state transition[33,35,36,43]. In our current high-resolution Cr-PSI-LHCI-LHCII structure, no CP29 is found and the proposed site for CP29 in the low-resolution projection map[36] is occupied by Lhca2 and/or Lhca9. A very low amount of CP29 was found in the preparation we used for the structural studies, and they may associate with PSI-LHCI in a small fraction of the particles as predicted by Drop et al.[36], but they were not further picked up for structural analysis in the present study.

In the two LHCII trimers associated with PSI-LHCI, the position of LHCII-1 is similar to that in higher plant PSI-LHCI-LHCII[30] with some shift and rotation. The phosphorylation of the N-terminal Thr residue of the LHCII-1 trimer in the stromal side upon state 1 to state 2 transition induces extensive contacts of LHCII-1 with the PsaO/PsaH/PsaL subunits, which stabilize its binding to the PSI core. However, the orientation of LHCII-1 trimer is rotated clockwise by 40° compared to that of LHCII in maize PSI-LHCI-LHCII, which may have been caused by its interactions with the nearby subunits different from that of maize PSI. One such nearby subunit is the LHCII-2 trimer associated with PSI through Lhca2 and LHCII-1. This is apparently different from the case in maize PSI-LHCI-LHCII[30], where only LHCII-1 interacts with the PsaO, PsaH and PsaL subunits from the PSI core. Multiple interactions between LHCII-1 and the PSI core and the LHCII-2 trimer suggests its rather tight binding, which may contribute to a larger antenna size or cross-section in the green alga that is likely required for state transition under low light conditions under water or in soil.

In summary, the Cr-PSI-LHCI-LHCII structure demonstrates a highly efficient network of EETs involved in the state transition of the green alga *C. reinhardtii*. Different from the PSI-LHCI-LHCII supercomplex of maize, the PSI-LHCI-LHCII in *C. reinhardtii* contains two LHCII trimers and the LHCII-1 has a different orientation compared to that of maize PSI-LHCI-LHCII, which resulted in some pathways of EET different from that of the maize. The movement and rotation of LHCII-1 enhances the interactions and connections between LHCII-1 and PSI core, which may increase the efficiency of EET from LHCII-1 to PSI core. On the other hand, the interactions of the LHCII-2 trimer with the PSI core is rather indirect, and most of them are mediated through LHCII-1 and Lhca2, making the binding of LHCII-2 trimer to the PSI core rather weaker. Some new pigments (Chls and carotenoids) were found in the interfacial region between LHCII and PSI. A low-energy Chl pair of PsaL is found and may mediate EET from LHCII to PSI, or be used as an energy trap to quench the excessive energy and prevent the PSI reaction center from being damaged. Some new carotenoids found in the PsaL, PsaH and PsaO subunits have a short distance with adjacent Chls and may play a vital role in quenching of harmful Chl triplet states under high light conditions. These features ensure a more efficient EET and photoprotection, which may be needed for green algae to adapt to the fluctuating light intensities during state transition in an aquatic environment.

## Methods

**Purification of the PSI-LHCI-LHCII.** *Chlamydomonas reinhardtii* cells (strain CC-137) were cultured in 10 liter of Tris-acetate-phosphate (TAP) liquid medium bubbled with 3–5% (v/v) $CO_2$-enriched air at 25 °C, under continuous white light illumination with an intensity of 30–50 μmol photons $m^{-2}$ $s^{-1}$. When the cells grow to the mid-logarithmic phase, they were cultured for an additional 12 h with an increased light intensity of 140 μmol photons $m^{-2}$ $s^{-1}$. Then the cells were harvested by centrifugation (4000 × g for 10 min, 25 °C) and resuspended in the TAP medium to a small volume of 200 ml. To induce the cells in state 1, the cultured cells were treated with 10 μM 3-(3,4-dichlorophenyl)-1,1-dimethylurea (an inhibitor of electron transfer from PSII to PQ pool) for 20 min under light illumination (30–50 μmol photons $m^{-2}$ $s^{-1}$) to oxidize the PQs[34–36]. Induction of state 2 in *C. reinhardtii* cells was carried out as follows[34–36]. The collected cells were incubated in a dark, anaerobic condition by purge with nitrogen in the presence of 100 mM sodium fluoride (NaF, an inhibitor of phosphatase). After 50 min, 1 mM sodium azide ($NaN_3$, an inhibitor of mitochondrial respiration) plus 7 μM cyanide p-(trifluoromethoxy)phenylhydrazone (FCCP, an uncoupling agent for depletion of ATP) was added, and the cells were incubated for 15 min. Subsequently, 100 mM glucose and 0.5 mg $ml^{-1}$ glucose oxidase (Sigma–Aldrich) were added, and the cells were incubated for another 20 min in the dark. Eventually, the nitrogen purge was removed and the treated cells were precipitated by centrifugation (4000 × g for 10 min, 25 °C) and used for thylakoid membrane preparation. Thylakoid membranes were isolated after disruption of the cells by a high-pressure chamber in a buffer of 0.3 M sucrose/25 mM HEPES-KOH (pH 7.5)/l mM $MgCl_2$ for the state 1 cells[44], and the same buffer supplemented with 1 mM $NaN_3$ and 10 mM NaF for the state 2 cells.

State transition was monitored by measuring the Chl fluorescence spectra of cells and thylakoid membranes at 77 K using a fluorescence spectrometer (F-4500, Hitachi, Japan) with an excitation wavelength of 436 nm.

PSI-LHCI-LHCII supercomplexes were purified as follows[34,36]. The thylakoid membranes were washed once with Buffer-1 (10 mM Hepes-KOH (pH 7.5), 5 mM EDTA, 1 mM $NaN_3$, 10 mM NaF) and the washed membranes were resuspended in Buffer-2 (20 mM Hepes-KOH (pH 7.5), 1 mM $NaN_3$, 10 mM NaF) to a Chl concentration of 1.2 mg $ml^{-1}$. The resuspended thylakoid membranes were solubilized with α-DDM by adding an equal volume of Buffer-2 containing 2.0% (w/v) α-DDM on ice for 10 min, and then the insoluble membranes were removed by centrifugation at 15,000 × g for 10 min. The solubilized membranes were loaded onto a continuous sucrose density gradient prepared by freeze-thawing of a 0.65 M sucrose solution containing 10 mM Hepes-KOH (pH 7.5), 1 mM $NaN_3$, 10 mM NaF, and 0.01% w/v α-DDM, and centrifuged at 256,800 × g for 18 h at 4 °C using a SW40Ti rotor with a Beckman Coulter Optima XPN-100 centrifuge. Two green bands in the middle position of the sucrose density gradient were collected carefully using a syringe and concentrated to a Chl concentration of 1 mg $ml^{-1}$ by using a 100 kDa cutoff membrane concentrator. Then, the sample was subjected to gel filtration chromatography (GE, Superose 6 Increase 10/300 GL) in a buffer containing 10 mM Hepes-KOH (pH 7.5), 50 mM NaCl, 1 mM $NaN_3$, 10 mM NaF, and 0.01% α-DDM. The peak fractions from the PSI-LHCI-LHCII band was collected for cryo-electron microscopic analysis.

**Characterization of the PSI-LHCI-LHCII.** The protein compositions of thylakoid membranes and purified PSI-LHCI and PSI-LHCI-LHCII supercomplexes were analyzed by electrophoresis using a gel containing 16% polyacrylamide and 7.5 M urea[45]. The amounts of sample loaded into each well were 20 μg Chls, and the gels were stained with Coomassie brilliant blue (CBB) R-250.

For mass spectrometry, CBB-stained bands were cut out from the gel, digested using sequencing grade, modified trypsin and the resultant peptides were extracted. The peptides were separated by an analytical column, which was a homemade, fused silica capillary column (75 μm ID, 150 mm length; Upchurch, Oak Harbor, WA) packed with C18 resin (300 Å, 5 μm; Varian, Lexington, MA), with a 60 min gradient elution at a flow rate of 0.30 μl/min with the EASY-nLC 1000 system, and was directly interfaced with the Thermo Orbitrap Fusion mass spectrometer. The MS/MS spectra from each LC-MS/MS run were searched against the selected database using Proteome Discovery searching algorithm (version 1.4).

UV absorption spectra were measured by a UV-Vis spectrophotometer (UV-2700, Shimadzu, Japan) at room temperature. Fluorescence emission spectra were measured at 77 K with a fluorescence spectrometer (F-4500, Hitachi, Japan) equipped with a xenon lamp source, and the spectra were recorded at a wavelength range from 600 to 800 nm with the excitation wavelength of 436 nm. The slit widths of both excitation and emission were set at 5.0 nm.

Pigment composition was analyzed by high-performance liquid chromatography (HPLC)[46]. The sample after size-exclusion chromatography was mixed with 90% (v/v) cold acetone to extract pigments, and the extract was centrifuged at 10,000 × g for 10 min under dim light. The supernatant was injected into a C18 reversed-phase column (Alltima$^{TM}$ C18 5 u) in a Waters e2695 separation module equipped with a Waters 2998 photodiode array detector. Pigments were identified based on their absorption spectra and elution times.

For phosphorylation analysis, western blotting was performed using an anti-pThr antibody[47] and Pro-Q Diamond Gel Staining. Proteins on the SDS-PAGE were transferred to a polyvinylidene fluoride membrane, and the membrane was probed with a pThr antibody (Cell Signaling Technology). All bands were visualized by enhanced chemiluminescence (Celvin Chemiluminescence Imaging System) after incubation with a horseradish peroxidase-conjugated secondary antibody (Goat Anti Mouse IgG (H + L)-HRP). For Pro-Q Diamond Gel Staining[48], the gels were stained with Pro-Q Diamond for 90 min and then decolored with the destaining solution containing 20% (v/v) acetonitrile and 50 mM sodium acetate (pH 4.0). The stained gels were imaged with a fluorescence detector (Model 2800, LI-COR).

**Sample preparation and cryo-EM data acquisition.** An aliquot of 2.5 μL purified PSI-LHCI-LHCII was applied to holey carbon grids covered with a graphene oxide film (Quantifoil R2/2, Au, 300 mesh), waited for 60 s, blotted for ~6.5 s at a humidity of 100% and 22 °C, and plunged into pre-cold liquid ethane with a Vitrobot (FEI). The FEI Titan Krios microscope was carefully aligned before data acquisition, including the coma-free alignment to minimize beam tilt. Cryo-EM

images of PSI-LHCI-LHCII were acquired using serialEM[49] with an accelerating voltage of 300 kV at a nominal magnification of ×22,500. The sample stage was tilted to 30° to surmount the preferred orientation problem of the supercomplex on the graphene oxide film. The dose rate of the electron beam was set to ~10 e⁻/s per physical pixel, and cryo-image stacks were recorded on a Gatan K2 summit camera using super-resolution at 4 frames/sec for 8.5 s. Defocus value on the specimen was set from −1.8 to −2.3 μm. A total of 5087 image stacks were collected. Drift correction of image stacks was performed with MotionCor2[50] to generate 2 × binned images with a calibrated pixel size of 1.307 Å/pixel on specimen. The total dose of each merged image is ~50 e⁻/Å².

**Image processing**. A total of 1,582,470 raw particle images were automatically picked with e2boxer of EMAN2[51], and the local defocus and astigmatism values of individual particles were determined using Gctf[52]. Image processing was accomplished with cryoSPARC[53]. After 2D classification, a total of 452,057 particles were selected for subsequent 3D classification, and 3 out of 7 classes with 283,763 particles were chosen. Refinements including non-uniform and local refinements were accomplished with a C1 symmetry, resulting in a structure at 3.42 Å resolution estimated by a cutoff value of 0.143 of the gold standard Fourier shell correlation curve[54] (Supplementary Fig. 2d). The local resolution was estimated using RELION[55].

**Model building and refinement**. The model of Cr-PSI-LHCI-LHCII was built by directly docking the structure of PSI-LHCI supercomplex from *C. reinhardtii* (PDB code: 6JO5) into the corresponding density map using UCSF Chimera[56]. The model of LHCII and PsaO were built with Coot using the structure of LHCII and PsaO from *C. reinhardtii* PSII-LHCII (PDB code: 6KAF) and maize PSI-LHCI-LHCII (PDB code: 5ZJI) as a reference, respectively[57]. The amino acid sequence of PsaO were then manually mutated to the corresponding sequence of PsaO from *C. reinhardtii*. The individual protein chains and amino acid residues as well as cofactors of the initial model of the Cr-PSI-LHCI-LHCII were further adjusted and processed manually according to the cryo-EM map with Coot[57]. Additionally, the protein subunits of the other two PSI-LHCI structures (PDB code: 6IJO and 6IGZ) from green algae (*C. reinhardtii* and *Bryopsis corticulans*) were then fitted into the density map of Cr-PSI-LHCI-LHCII to optimize the structural model. The structures of each subunit were then refined in real space using Phenix[58] and displayed with UCSF Chimera and PyMOL (Molecular Graphics System, LLC).

**Reporting summary**. Further information on research design is available in the Nature Research Reporting Summary linked to this article.

## Data availability
The cryo-EM density map and atomic models have been deposited in the Electron Microscopy Data Bank and the Protein Data Bank (EMD ID code 30536 and PDB ID code 7D0J). The data that support the findings of this study are available from the corresponding authors upon reasonable request. Source data are provided with this paper.

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

## Acknowledgements

We thank Shenghai Chang in the Center of Cryo-Electron Microscopy (CCEM), Zhejiang University for the technical assistance on cryo-EM data collection, C. Ma from the Protein facility, School of Medicine, Zhejiang University for providing this platform for sample purification, X. Meng in Center of Biomedical Analysis, Tsinghua University, for protein MS analysis, L. Yu from the Institute of Botany, CAS for assistance in model building, W. Tang and Y. Yin from the Institute of Botany, CAS for instrumental supports in sample preparation, fluorescence measurement and high-performance liquid chromatography analysis. The project was funded by the National Key R&D Program of China (2017YFA0503700, 2017YFA0504803, 2018YFA0507700, 2020YFA0907600, 2019YFA0906300), a Strategic Priority Research Program of CAS (XDB17000000, XDA27050402), the Chinese Academy of Sciences Key Research Program of Frontier Sciences (QYZDY-SSW-SMC003), the Fundamental Research Funds for the Central Universities (2018XZZX001-13), and Youth Innovation Promotion Association of CAS (2020081).

## Author contributions

G.H., X.Z., and J.-R.S. conceived the project, L.S., G.H., and Z.M. performed the sample preparation and characterization; Z.H. and X.Y. collected the cryo-EM data; X.Z. processed the cryo-EM data and reconstructed the cryo-EM map; L.S. built the structure model; W.W. refined the structure; L.S. and G.H. analyzed the structure; L.S., G.H., Z.H., X.Z., and J.-R.S. jointly wrote the manuscript. All authors discussed and commented on the results and the manuscript.

## Competing interests

The authors declare no competing interests.
