## [Peer Review File · Nature Communications]

REVIEWER COMMENTS

Reviewer #1 (Remarks to the Author):

Huang, Shen et al. report the high-resolution structure of the PSI-LHCI-LHCII supercomplex, representing the PSI antenna composition when the cells are in "state II" where both LHCI and LHCII subunits are bound. A similar structure has previously been solved from maize, a higher plant, but never from a green algae, which in this case is *C. reinhardtii*. This structure from a green algae is important because these organisms are aquatic unlike maize, and thus experience different light conditions, which results in unique supercomplex organization, thus providing insight into the diversity of light-harvesting in nature.

The manuscript is very nicely written, providing a succinct and informative introduction, and a detailed description of the characterization carried out to ensure the expected spectral properties of the complex containing LHCII and even phosphorylation of LHCII subunits. The authors then provide a good large-scale comparison with other structures, followed by observations made in finer detail that identify the roles of specific residues, groups of residues, and pigments, and then a good presentation of possible energy transfer pathways in the complex.

The *C. reinhardtii* PSI-LHCI-LHCII supercomplex exhibits similar subunit and pigment composition relative to the *C. reinhardtii* PSI-LHCI supercomplex that has previously been solved except the addition of the two LHCII trimers. One of the LHCII trimers lies in a similar location to that observed from the maize PSI-LHCI-LHCII structure, and the other further away from the core, toward Lhca2. To me, the most noteworthy small-scale findings are that LHCII-1 is bound somewhat differently than that from maize, which is probably due to the addition of the nearby LHCII-2 which is unique to *C. reinhardtii*. LHCII-2, however, exhibits relatively few possible energy transfer pathways, that probably occur through either LHCII-1 and/or Lhca2. PsaO, PsaH, and some nearby pigments are newly-identified which is probably due to stabilization by the LHCIIs and a good description of these are provided. Also, the authors show the specific phosphorylated residue's cryo-EM density assigned to the LHCBM3 subunit of LHCII that is implicated in state 2 transition.

I support publication of this article in Nature Communications, but I suggest the following edits:

- 1) There are many Chl b assignments in the structure, but at the present resolution Chl a and Chl b cannot be distinguished. Thus, I assume these assignments were made based off homology with structures where the achieved resolution was high enough to distinguish Chl b. Please describe how the Chl b sites were assigned, especially the rationale for why these assignments are confident. Conversely, if the Chl b cannot be confidently assigned, such transparency would also be good to state.
- 2) Line 29: Typo – "large" should be "larger".
- 3) Line 89: The authors direct the reader to "Materials and Methods", but the corresponding section is named only "Methods".
- 4) Line 138: Do the authors mean "cryo-EM structure of green algae PSI-LHCI"?
- 5) Line 358: "High" should be "higher".
- 6) Discussion: Throughout the discussion, the authors compare the *C. reinhardtii* PSI-LHCI-LHCII supercomplex to that from "higher plant". If the authors mean a single higher plant (e.g. *Z. mays*), they should be specific. If the authors mean all higher plants, they should pluralize "plant" to "plants".
- 7) Line 408: "Under the" should be "in an".

Reviewer #2 (Remarks to the Author):

In the linear mode of photosynthetic electron transfer the two photosystems act in series, and for

optimal photosynthesis they should work at similar rates. These rates are balanced under low light by state transitions, which reflect a regulatory feedback loop that responds to the redox poise of the chain and dynamically allocates part of the LHCII antenna to either PSII or PSI.

Phosphorylation of LHCII is a key factor in state transitions, promoting association of part of LHCII to PSI in state 2. The green alga *Chlamydomonas* shows state transitions of large amplitude, which is reflected by the fact that the PSI-LHCI-LHCII complex that forms in state 2 contains two LHCII trimers, whereas the corresponding plant complex contains only one. While the overall arrangement of PSI-LHCI-LHCII in *Chlamydomonas* was previously determined, the new structure presented here offers much better resolution and allows a precise comparison with the recently determined structure of the corresponding complex from maize. Furthermore, compared to previous structures of PSI-LHCI, the new structure offers novel information on several subunits that were not resolved previously, and maps pigment molecules that had not been identified. Because the LHCII protein family has evolutionarily diverged and is more complex in algae than in plants, this work is also interesting for comparison of the algal LHCs with the plant Lhcs.

Comments:

1. The authors identify phosphorylated LHCBM1 as a key determinant of PSI-LHCI-LHCII formation, similar to P-Lhcb2 in plants. However comparison of the Coomassie and the Pro-Q stains in Fig S2c suggests that LHCBM3 (Type I) may be more heavily phosphorylated than LHCBM1, although present in lower amounts. Is phosphorylation of LHCs, and in particular of LHCBM3, apparent anywhere else in the two LHCII trimers in the structure? Is the structure well-enough resolved to allow assignment of specific LHCs to specific positions in the two trimers? Or conversely does the data favor a heterogeneous arrangement of the diverse isoforms, with only P-LHCBM1 specifically located?

2. It would be interesting to directly compare the molecular recognition of pThr of LHCBM1 in LHCII-1 by PSI (Fig 3) with the similar recognition of pLhcb2 in Maize (Pan et al, 2018: Fig 2E). How evolutionarily conserved is this binding site?

3. In *Chlamydomonas*, LHC4 (CP29) was previously reported to be part of the mobile antenna that associates with PSI in State 2. However the new structure presented here does not include this monomeric antenna, which is present in only very low amounts in the preparation. Furthermore, the site proposed for LHC4 in the low-resolution structure (Drop et al, 2014) seems in fact occupied by Lhca2 and/or Lhca9 in the new one. The authors should more explicitly comment about this in the Discussion (line 372).

4. The authors make repeated mentions of energy dissipation triggered by high light in PSI (lines 212, 346, 400, 743). Is there evidence for such quenching involving carotenoids in PSI? Would such a quenching mechanism be constitutively present also in the relatively low light that was used to grow the cells (at competitive disadvantage for the alga)? Is this related to the disputed suggestion that in St2 part of the LHCII that detaches from PSII does not connect to PSI and seems quenched (Nawrocki PMID: 27249564; Unlu PMID: 24550508; Nagy PMID: 24639515)? Or could this be related to the controversial role of Zea reported in plants (Ballotari PMID: 24872450; Tian PMID: 28416696)? Since the algae were grown in moderate light plus acetate, presumably LHCSR3 and LHCSR1 were not present, and they were not found in the complex. Nevertheless could LHCSR-mediated quenching induce a conformational change that activates dissipation involving the carotenoids (Allorent PMID: 23424243; Kosuge PMID: 29555769) and could the supercomplex have artefactually adopted this configuration during purification?

5. Line 122. Antheraxanthin is present both in PSI-LHCI from St1 and in PSI-LHCI-LHCII from St2. Why is a xanthophyll cycle in St2 proposed? Was this pigment identified in the structure of the supercomplex and if so where?

6. Line 66. This should be better explained, because (a) only part of LHCII is phosphorylated and associates to PSI, and (b) phosphorylation of Lhcb2 is most relevant for state transitions, not

phosphorylation of Lhcb1 (Pan et al. 2018; Crepin PMID: 26392145 ; Longoni PMID: 26438789).

7. Lines 356 and 385. The laboratory strain of *Chlamydomonas* was isolated from soil, not from open water.

8. Minor points: line 29: larger; line 54: and -> that; line 349: Discussion; line 366: Arabidopsis

Responses to the comments of Reviewer #1

Comments

Huang, Shen et al. report the high-resolution structure of the PSI-LHCI-LHCII supercomplex, representing the PSI antenna composition when the cells are in “state II” where both LHCI and LHCII subunits are bound. A similar structure has previously been solved from maize, a higher plant, but never from a green algae, which in this case is *C. reinhardtii*. This structure from a green algae is important because these organisms are aquatic unlike maize, and thus experience different light conditions, which results in unique supercomplex organization, thus providing insight into the diversity of light-harvesting in nature.

The manuscript is very nicely written, providing a succinct and informative introduction, and a detailed description of the characterization carried out to ensure the expected spectral properties of the complex containing LHCII and even phosphorylation of LHCII subunits. The authors then provide a good large-scale comparison with other structures, followed by observations made in finer detail that identify the roles of specific residues, groups of residues, and pigments, and then a good presentation of possible energy transfer pathways in the complex.

The *C. reinhardtii* PSI-LHCI-LHCII supercomplex exhibits similar subunit and pigment composition relative to the *C. reinhardtii* PSI-LHCI supercomplex that has previously been solved except the addition of the two LHCII trimers. One of the LHCII trimers lies in a similar location to that observed from the maize PSI-LHCI-LHCII structure, and the other further away from the core, toward Lhca2. To me, the most noteworthy small-scale findings are that LHCII-1 is bound somewhat differently than that from maize, which is probably due to the addition of the nearby LHCII-2 which is unique to *C. reinhardtii*. LHCII-2, however, exhibits relatively few possible energy transfer pathways, that probably occur through either LHCII-1 and/or Lhca2. PsaO, PsaH, and some nearby pigments are newly-identified which is probably due to stabilization by the LHCIIs and a good description of these are provided. Also, the authors show the specific phosphorylated residue’s cryo-EM density assigned to the LHCBM3 subunit of LHCII that is implicated in state 2 transition.

I support publication of this article in Nature Communications, but I suggest the following edits:

Author answers:

We greatly appreciate the reviewer for his/her highly positive and encouraging comments. We have carefully considered the comments raised by the reviewer and revised our manuscript accordingly, which are listed below.

1) There are many Chl b assignments in the structure, but at the present resolution Chl a and Chl b cannot be distinguished. Thus, I assume these assignments were made based off homology with structures where the achieved resolution was high enough to

distinguish Chl *b*. Please describe how the Chl *b* sites were assigned, especially the rationale for why these assignments are confident. Conversely, if the Chl *b* cannot be confidently assigned, such transparency would also be good to state.

Answers:

Yes, you are right that at the current resolution, we were not able to discriminate between Chl *a* and Chl *b* completely. We have modeled the site of Chl *b* based on the structure of PSI-LHCI and isolated LHCI⁸ determined at higher resolution previously, and added the following sentence to indicate this in the revised manuscript (lines 163-165, page 7).

“Due to the limited resolution, it was not possible to distinguish between Chl *a* and Chl *b* in most cases. Thus, we modeled these pigments according to the high resolution structures of Cr-PSI-LHCI¹³ and isolated LHCI⁸ reported previously.”

2) Line 29: Typo - “large” should be “larger”.

Answers:

We have corrected “large” to “larger” in the revised manuscript. Thank you.

3) Line 89: The authors direct the reader to “Materials and Methods”, but the corresponding section is named only “Methods”.

Answers:

We have corrected “Materials and Methods” to “Methods” in the revised manuscript. Thank you.

4) Line 138: Do the authors mean “cryo-EM structure of green algae PSI-LHCI”?

Answers:

Yes, we mean “cryo-EM structure of green algae PSI-LHCI”. We have added “PSI-LHCI” after the word “algae” in the revised manuscript. Thank you.

5) Line 358: “High” should be “higher”.

Answers:

We have corrected “High” to “higher” in the revised manuscript. Thank you.

6) Discussion: Throughout the discussion, the authors compare the *C. reinhardtii* PSI-LHCI-LHCI⁸ supercomplex to that from “higher plant”. If the authors mean a single higher plant (e.g. *Z. mays*), they should be specific. If the authors mean all higher plants, they should pluralize “plant” to “plants”.

Answers:

We mean a single higher plant, and have replaced the “higher plant” with “maize” in most cases, and changed “higher plant” to “higher plants” in a few cases where we wanted to say.

7) Line 408: “Under the” should be “in an”.

Answers:

We have corrected “Under the” to “in an” in the revised manuscript. Thank you.

Responses to the comments of Reviewer #2

Comments

In the linear mode of photosynthetic electron transfer the two photosystems act in series, and for optimal photosynthesis they should work at similar rates. These rates are balanced under low light by state transitions, which reflect a regulatory feedback loop that responds to the redox poise of the chain and dynamically allocates part of the LHCII antenna to either PSII or PSI. Phosphorylation of LHCII is a key factor in state transitions, promoting association of part of LHCII to PSI in state 2. The green alga *Chlamydomonas* shows state transitions of large amplitude, which is reflected by the fact that the PSI-LHCI-LHCII complex that forms in state 2 contains two LHCII trimers, whereas the corresponding plant complex contains only one. While the overall arrangement of PSI-LHCI-LHCII in *Chlamydomonas* was previously determined, the new structure presented here offers much better resolution and allows a precise comparison with the recently determined structure of the corresponding complex from maize. Furthermore, compared to previous structures of PSI-LHCI, the new structure offers novel information on several subunits that were not resolved previously, and maps pigment molecules that had not been identified. Because the LHCII protein family has evolutionarily diverged and is more complex in algae than in plants, this work is also interesting for comparison of the algal LHCBMs with the plant Lhcbs.

Answers:

We greatly appreciate the reviewer for his/her highly positive and encouraging comments. We have carefully considered the comments raised by the reviewer and modified our manuscript accordingly, which are listed below.

1. The authors identify phosphorylated LHCBM1 as a key determinant of PSI-LHCI-LHCII formation, similar to P-Lhcb2 in plants. However comparison of the Coomassie and the Pro-Q stains in Fig S2c suggests that LHCBM3 (Type I) may be more heavily phosphorylated than LHCBM1, although present in lower amounts. Is phosphorylation of LHCBMs, and in particular of LHCBM3, apparent anywhere else in the two LHCII trimers in the structure? Is the structure well-enough resolved to allow assignment of specific LHCBMs to specific positions in the two trimers? Or conversely does the data favor a heterogeneous arrangement of the diverse isoforms, with only P-LHCBM1 specifically located?

Answers:

The results presented in this study clearly indicated that both LHCBM1 (Type IV) and LHCBM3 (Type I) are phosphorylated, and LHCBM3 is even more heavily phosphorylated than LHCBM1. From the structure, only the phosphorylated terminal

of the LHCII-1 isoform was identified to be involved in the direct interaction with the PSI-LHCI moiety. This was based on the sequence difference of LHCBM1 (Type IV) and LHCBM3 (Type I) and the cryo-EM density we obtained. We were not able to assign pLHCBM3 to a specific position of LHCII. From our current data, it seems unlikely that the pLHCBM3 is present in any of the two LHCII trimers attached to PSI-LHCI, but further studies may resolve this issue.

2. It would be interesting to directly compare the molecular recognition of pThr of LHCBM1 in LHCII-1 by PSI (Fig 3) with the similar recognition of pLhcb2 in Maize (Pan et al, 2018: Fig 2E). How evolutionarily conserved is this binding site?

Answers:

This is a very good comment. According to the reviewer's comment, we compared the molecular recognition of pThr of LHCBM1 in LHCII-1 by PSI with the similar recognition of pLhcb2 in Maize (Pan et al, 2018: Fig 2E), which shows that this binding site is evolutionarily conserved between the green alga *C. reinhardtii* and the high plant maize. We added this information in the revised text (see below, lines 264-267) and a new Supplementary Fig. 10 into the revised manuscript, respectively.

“It is interesting to note that the binding site of pThr is evolutionarily and structurally conserved in both green alga *C. reinhardtii* and the high plant maize PSI-LHCI-LHCII (Supplementary Fig. 10).”

3. In *Chlamydomonas*, LHCB4 (CP29) was previously reported to be part of the mobile antenna that associates with PSI in State 2. However the new structure presented here does not include this monomeric antenna, which is present in only very low amounts in the preparation. Furthermore, the site proposed for LHCB4 in the low-resolution structure (Drop et al, 2014) seems in fact occupied by Lhca2 and/or Lhca9 in the new one. The authors should more explicitly comment about this in the Discussion (line 372).

Answers:

As the reviewer pointed out, CP29 was previously reported to be part of the mobile antenna that associates with PSI in the State 2 cells of *C. reinhardtii*, and the amounts varied with the methods of preparation. In the present study, the amount of CP29 was very low in our preparation used for cryo-EM analysis, and no CP29 was found to be associated with PSI-LHCI in state 2. Instead, the site proposed for CP29 binding in the low-resolution structure (Drop et al, 2014) is occupied by Lhca2 and/or Lhca9 in our structure. This is consistent with the very low amount of CP29 in the Cr-PSI-PSII preparation of our study. This low amount of CP29 may associate with PSI-LHCI in a

small fraction of the particles that were not picked up for the structural analysis in the current study.

Based on these analyses, we added the following sentences in the discussion section of the revised manuscript to explain it more explicitly (lines 384-389, revised manuscript).

“In our current high-resolution Cr-PSI-LHCI-LHCII structure, no CP29 is found and the proposed site for CP29 in the low-resolution projection map³⁶ is occupied by Lhca2 and/or Lhca9. A very low amount of CP29 was found in the preparation we used for the structural studies, and they may associate with PSI-LHCI in a small fraction of the particles as predicted by Drop et al.³⁶, but they were not further picked up for structural analysis in the present study.”

4. The authors make repeated mentions of energy dissipation triggered by high light in PSI (lines 212, 346, 400, 743). Is there evidence for such quenching involving carotenes in PSI? Would such a quenching mechanism be constitutively present also in the relatively low light that was used to grow the cells (at competitive disadvantage for the alga)? Is this related to the disputed suggestion that in St2 part of the LHCII that detaches from PSII does not connect to PSI and seems quenched (Nawrocki PMID: 27249564; Unlu PMID: 24550508; Nagy PMID: 24639515) ? Or could this be related to the controversial role of Zea reported in plants (Ballotari PMID: 24872450; Tian PMID: 28416696)? Since the algae were grown in moderate light plus acetate, presumably LHCSR3 and LHCSR1 were not present, and they were not found in the complex. Nevertheless could LHCSR-mediated quenching induce a conformational change that activates dissipation involving the carotenes (Allorent PMID: 23424243; Kosuge PMID: 29555769) and could the supercomplex have artefactually adopted this configuration during purification?

Answers:

We thank the reviewer for these comments and information on this issue. Since these carotenes are newly found carotenoids in this study, there is no evidence and study for such carotenes to be involved in quenching in PSI. It is the same case in the low light condition. Thus, we are not able to give a conclusive answer on whether it could be related to the controversial role of Zea reported in plants (Ballotari PMID: 24872450; Tian PMID: 28416696).

There is indeed no LHCSR1 or LHCSR3 in the current PSI-PSI-LHCII complex, so the possibility of a LHCSR-mediated conformational changes of these carotenes is rather small. In terms of the possibility of an artifactually adopted configuration of carotenes during purification, we cannot exclude this possibility, but we think this is

very small because the pigments are located inside the protein subunits and no detergent molecules were found around these carotenes.

Since the carotenes mentioned in the manuscript are found in the PsaH/PsaL/PsaO subunits of PSI-LHCI, in our opinion, it should not be related to the disputed suggestion that in St2 part of the LHCII that detaches from PSII does not connect to PSI and seems quenched (Nawrocki PMID: 27249564; Unlu PMID: 24550508; Nagy PMID: 24639515).

Usually, the carotenoids may have two major functions in photosynthesis (Cogdell, R J. Pure Applied Chem, 1985, 57, 723-728), namely light capture and photoprotection by quenching harmful chlorophyll (Chl) triplet states to prevent the formation of highly toxic singlet oxygen. Chl triplet quenching proceeds by intersystem crossing and requires carotenoids to be located close to the porphyrin head group of Chl (Standdfuss et al., The EMBO Journal, 2005, 24, 919-928.). Therefore, according to the reviewer's comments, we replaced the word "energy dissipation" or "energy quenching" with "photoprotection" in the revised manuscript.

5. Line 122. Antheraxanthin is present both in PSI-LHCI from St1 and in PSI-LHCI-LHCII from St2. Why is a xanthophyll cycle in St2 proposed? Was this pigment identified in the structure of the supercomplex and if so where?

Answers:

As described in the text, the PSI-LHCI sample in which the pigment of antheraxanthin was found was also obtained from the state 2 cells instead of state 1 cells. Since this pigment was not detected in PSI-LHCI from the state 1 cells of this green alga previously, we specifically proposed the xanthophyll cycle in state 2. We added this information in the legend of revised Supplementary Fig1, and added the following sentence in the revised manuscript to explain this (lines 126-127).

"Since this pigment was not found in PSI-LHCI isolated from state 1 cells of *C. reinhardtii*¹², this indicates that the xanthophyll cycle may occur ...".

However, at the current resolution, we were not able to discriminate between different types of carotenoids (neoxanthin, lutein, violaxanthin, antheraxanthin, and β -carotene), so the pigment antheraxanthin was not able to be identified in the structure.

6. Line 66. This should be better explained, because (a) only part of LCHII is phosphorylated and associates to PSI, and (b) phosphorylation of Lhcb2 is most relevant for state transitions, not phosphorylation of Lhcb1 (Pan et al. 2018; Crepin

PMID: 26392145 ; Longoni PMID: 26438789).

Answers:

We agree with the reviewer's comments and have added the following explanations into the revised manuscript (lines 69-70). We also cited these references in the text.

“Recent reports show that one of the LHCII isoform, Lhcb2 is in its phosphorylated form in higher plants in state 2²⁸⁻³⁰.”

7. Lines 356 and 385. The laboratory strain of *Chlamydomonas* was isolated from soil, not from open water.

Answers:

The reviewer is right, and according to the reviewer's comment, we added words “and in soil” in the revised manuscript.

8. Minor points: line 29: larger; line 54: and -> that; line 349: Discussion; line 366: Arabidopsis

Answers:

We have corrected the misspelled words and errors in the revised text. Thank you.

REVIEWER COMMENTS

Reviewer #1 (Remarks to the Author):

The authors have adequately addressed my comments and I recommend this manuscript for publication in its current form.

Reviewer #2 (Remarks to the Author):

In their revision, the authors have addressed all comments and made corresponding modifications to the text. They have also added a new supplemental figure (S10) to compare the molecular details of the interaction of the pThr site of LHCII with PSI in *Chlamydomonas* and Maize, showing its remarkable conservation.

I have only one remaining concern, and that is about LHCBM1 and LHCBM3. Looking into this again, it occurred to me that the npq5 mutant of *Chlamydomonas*, which is deficient for LHCBM1, and also an anti-sense line of LHCBM1 were both found to be still capable of state transitions (and were deficient in qE) (Elrad et al., PMID: 12172023; Ferrante et al, PMID: 22431727). It is thus puzzling that pLHCBM1 should be key to the formation of the PSI-LHCI-LHCII complex as described here, which seems to contradict the previous genetic analysis. LHCBM3 belongs to type I, together with LHCBM 4/6/8, so they would not have been distinguished in gel electrophoresis (Fig 2). The sequence around pThr in LHCBM 4/6/8 (...KKpTA...) is somewhat similar to that of LHCBM1 (...RRpTV...), and from Figures 9 and 10 it is difficult to judge whether LHCBM4/6/8 could also fit the structural data at the binding site, and also in other parts of the trimer. Therefore I would like to ask the authors to carefully consider whether LHCBM4/6/8 could fit in their structure in place of LHCBM1.

Regarding point 6, the new statement is not quite accurate. Both Lhcb1 and Lhcb2 are heavily phosphorylated in State 2, but only pLhcb2 is present in the trimer bound to PSI.

Responses to the remarks of Reviewer #2

Remarks

In their revision, the authors have addressed all comments and made corresponding modifications to the text. They have also added a new supplemental figure (S10) to compare the molecular details of the interaction of the pThr site of LHCII with PSI in *Chlamydomonas* and Maize, showing its remarkable conservation.

I have only one remaining concern, and that is about LHCBM1 and LHCBM3. Looking into this again, it occurred to me that the npq5 mutant of *Chlamydomonas*, which is deficient for LHCBM1, and also an anti-sense line of LHCBM1 were both found to be still capable of state transitions (and were deficient in qE) (Elrad et al., PMID: 12172023; Ferrante et al, PMID: 22431727). It is thus puzzling that pLHCBM1 should be key to the formation of the PSI-LHCI-LHCII complex as described here, which seems to contradict the previous genetic analysis. LHCBM3 belongs to type I, together with LHCBM 4/6/8, so they would not have been distinguished in gel electrophoresis (Fig 2). The sequence around pThr in LHCBM 4/6/8 (...KKpTA...) is somewhat similar to that of LHCBM1 (...RRpTV...), and from Figures 9 and 10 it is difficult to judge whether LHCBM4/6/8 could also fit the structural data at the binding site, and also in other parts of the trimer. Therefore I would like to ask the authors to carefully consider whether LHCBM4/6/8 could fit in their structure in place of LHCBM1.

Author answers:

We agree with the reviewer's remarks, and carefully compared the sequences and cryo-EM map around pThr of the N-termini of LHCBM1 and LHCBM3/4/8. We can clearly exclude the possibility of LHCBM3 as the phosphorylated LHCBM subunit involved in the interactions with the PSI core subunits based on the cryo-EM density. However, the N-termini of LHCBM1 (RRTV) and LHCBM4/8/6 (KKTA) are rather similar, and we were not able to distinguish them based on the current cryo-EM map. We carefully checked the Coomassie bands identified based on the mass spectrometry analysis, and find the presence of a small amount of LHCBM4 and LHCBM8, but no LHCBM6 in the Type I LHCBMs. The amount of LHCBM4 is relatively lower than that of LHCBM3, but much higher than that of LHCBM8.

Based on the higher abundance of LHCBM4 compared to LHCBM8, and the fact that deletion of LHCBM1 does not inhibit the state transition of *C. reinhardtii* as reported previously (Elrad et al., PMID: 12172023; Ferrante et al, PMID: 22431727), we tentatively assigned the phosphorylated LHCBM to be LHCBM4. We have revised this part of the text as following, in which we also added a sentence to indicate that there is a possibility that the phosphorylated LHCBM is LHCBM1, and deletion of this LHCBM1 may induce phosphorylation and movement of other LHCBM to PSI. We also added the two references indicated by the reviewer, and

revised the corresponding Figures 2, 5 and Supplementary Figures 9, 10.

“The cryo-EM density around the N-terminal tail of monomer-1 of LHCII-1 matched with the conserved sequence “RRTV” of LHCBM1 as well as “KKTA” of LHCBM4/8, but not with the N-terminal “GKTA” sequence of LHCBM3, where the residue Thr is phosphorylated. Based on the higher abundance of LHCBM4 compared to LHCBM8, and the fact that deletion of LHCBM1 does not inhibit the state transition of *C. reinhardtii* as reported previously^{40,41}, we tentatively assign phosphorylated LHCBM4 to the LHCII-1 subunit involved in the interactions with the PSI core subunits. The cryo-EM density of the N-terminal region can also be fitted with the N-terminal sequence of LHCBM1 (RRTV). If this is the case, there is a possibility that deletion of LHCBM1 may cause other LHCBMs to be phosphorylated and moved to PSI under state transition conditions.” (lines 259-268 of revised manuscript).

Regarding point 6, the new statement is not quite accurate. Both Lhcb1 and Lhcb2 are heavily phosphorylated in State 2, but only pLhcb2 is present in the trimer bound to PSI.

Author answers:

We agree with the reviewer’s remarks, and have modified it as suggested by the reviewer in the revised manuscript (lines 69-71, revised manuscript). Thank you.